# Segregation of endoderm and mesoderm germ layer identities in the diploblast *Nematostella vectensis*

Emmanuel Haillot[1], Tatiana Lebedeva [1], Julia Steger[1], Grigory Genikhovich [1], Juan D. Montenegro [1], Alison G. Cole [1] & Ulrich Technau [1,2] ✉

A recent study suggested that morphologically diploblastic sea anemones (Cnidaria) have three segregated germ layer identities corresponding to the bilaterian germ layers. Here, we investigated how these germ layer identities are specified during early development of the sea anemone *Nematostella vectensis*. Our gene expression analysis shows that the mesodermal territory is specified at the animal pole at 6 h postfertilization, followed by the specification of the definitive endoderm between mesoderm and ectoderm. We then assessed the role of β-catenin, MAPK, and Notch signaling during mesoderm and endoderm formation. We show that the mesodermal marker genes are activated by MAPK signaling while being repressed elsewhere by β-catenin signaling. Delta-expressing mesoderm then signals to Notch-expressing ectoderm, inducing the definitive endoderm domain at the mesoderm/ectoderm interface. Gain- and loss-of-function experiments showed that Notch signaling is sufficient for endoderm induction. Based on our results, we propose a model of germ layer specification in *Nematostella* defined by a crosstalk of MAPK, β-catenin, and Notch signaling. Given the similarity of the germ layer specification between the sea anemone and echinoderms, we propose that triploblasty may have predated the split of cnidarians and bilaterians.

All bilaterians are composed of three embryonic germ layers, ectoderm, endoderm, and mesoderm, which give rise to all tissues in the adult organism. In contrast, Cnidaria, the sister group of Bilateria, consists of only two cell layers. Traditionally, it was assumed that the two cell layers of cnidarians are homologous to bilaterian endoderm and ectoderm, and that the mesoderm arose at the base of Bilateria. To trace the evolutionary origin of mesoderm, conserved transcription factors involved in mesoderm formation and differentiation in Bilateria were cloned from various cnidarians[1–4]. Most of these genes are expressed in the inner cell layer, which led to the suggestion that the inner layer constitutes an endomesoderm, an evolutionary precursor of mesoderm and endoderm. However, expression analyses of conserved endodermal and mesodermal markers throughout development of the sea anemone *Nematostella vectensis* challenged this view and showed that the inner layer has a molecular profile reminiscent of the bilaterian mesoderm, and gives rise to muscles, gonads, and nutrient storage, typical derivatives of mesoderm in Bilateria[5]. Conversely, the pharyngeal ectoderm and its derivatives, the septal filaments at the distal tips of the mesenteries, express many conserved endodermal markers and differentiate into digestive gland cells and insulin-secreting cells, characteristic of endoderm derivatives in Bilateria[5]. These findings have challenged the presumed homology of cnidarian inner and outer cell layers with ectoderm and endoderm.

In Bilateria, endomesoderm specification and its subsequent subdivision into endo- and mesoderm is one of the first cell fate decisions and hence a key step in animal development. In ambulacrarian deuterostomes, such as echinoderms and hemichordates, in urochordates, and at least in some spiralian protostomes such as

[1]Department of Neurosciences and Developmental Biology, Faculty of Life Sciences, University of Vienna, Djerassiplatz 1, 1030 Vienna, Austria. [2]Research Platform SINCEREST, University of Vienna, Djerassiplatz 1, 1030 Vienna, Austria. ✉e-mail: ulrich.technau@univie.ac.at

nemertines, endomesoderm formation is governed by β-catenin signaling[6–9]. In echinoderms, Notch signaling then segregates endomesoderm into endoderm and mesoderm[8,10,11], while mitogen-activated protein kinase (MAPK) signaling is involved in the specification of the mesoderm. The pro-mesodermal function of MAPK has also been reported in ecdysozoans and spiralians, suggesting an ancestral role in the common bilaterian ancestor[12]. Although the regulatory relationships between these three signaling pathways may vary, their involvement in the bilaterian germ layer specification is a recurrent theme.

In this work, we investigate the roles of MAPK, Notch, and β-catenin signaling pathways in the germ layer specification in the sea anemone *Nematostella vectensis* to understand how germ layers are specified in a member of Cnidaria, the sister clade of bilaterians and to test the hypothesis of potential cnidarian triploblasty. We show that the mutually antagonistic actions of MAPK and β-catenin signaling first segregate the mesodermal and ectodermal germ layer identities. Then, at the boundary between these layers, activation of Notch signaling specifies the endodermal germ layer identity. These intricate signaling interactions as well as downstream transcription factor activation, are highly reminiscent of germ layer segregation in the sea urchin, a deuterostome, raising the possibility of a common evolutionary origin.

## Results

### Mesoderm and ectoderm segregate before the endoderm

*Nematostella* eggs contain ubiquitously distributed maternal mRNAs of genes whose zygotic expression later marks aboral ectoderm and becomes gradually cleared first from the oral hemisphere of the embryo[13]. Previous studies identified a number of specific marker genes for endoderm or mesoderm[1,2,5,14]. To determine the precise timing of mesoderm and endoderm initiation during development and to define more clearly the key steps responsible for their emergence, we generated single cell RNA libraries of 8, 10 and 12 hours postfertilisation (hpf). Additionally, single-cell RNAseq data previously generated by our lab at 18, and 24 hpf[15,16] were also used for this analysis. After cell clustering, we were able to distinguish the cell populations with ectodermal, mesodermal and endodermal identities. Notably, at 8 hpf, only clusters of mesodermal and ectodermal cells are clearly identifiable (Supplementary Fig. 1). The ectodermal cells already express *brachyury*, a gene considered an endodermal marker at 12 hpf. This earlier expression corresponds to the future oral midbody ectoderm and will be maintained until gastrula. Therefore, at an early stage, *brachyury* is an endodermal marker as well as a marker of the oral midbody ectoderm. Our data show sequential activation of mesodermal genes (Fig. 1A). The first expression of zygotic genes encoding transcription factors (TFs) such as *tbx19-like, gsc2-like, duxABC1* and *duxABC2* starts around 6 hpf. By 8 hpf, *pitx1-like, fgfa1,* and *erg* are turned on in addition (Fig. 1A, D). This is followed by the expression of *nkx2.2B* detectable at 10 hpf and by the expression of another set of mesodermal genes, such as *zicA, snailA, mitf-like, runx, six4,* and *smad1/5* by 12hpf. At 14hpf, *hand2, zc3h12-like, nkx2.2D, otxB,* and at 16hpf, *isx-like 2,* and *otxC* become detectable. At 18 hpf, the expression of *hmx2* starts, shortly before the onset of gastrulation (Fig. 1B,D). The ectodermal cells express genes encoding homeobox transcription factors such as *koza-like1/2,* or a bHLH TF-like in a pattern complementary to the mesodermal gene expression (Fig. 1A). The expression of the future definitive endoderm markers *brachyury, foxA, wnt1,* and *wnt3* starts to be detected at 8 hpf (*brachyury*), 10 hpf *(wnt1* and *wnt3*), and 12 hpf *(foxA)* (Fig. 1C)[17]. *Brachyury, wnt1* and *wnt3* start to be expressed in the cells expressing mesodermal markers[18], but by 12 hpf *brachyury* expression shifts into a ring of cells surrounding the mesoderm followed by *wnt1* and *wnt3* gene expression (Fig. 1C). From 12 hpf until 20hpf, endodermal markers, *foxA, wnt1* and *wnt3* are expressed in a one cell diameter-wide ring surrounding the mesoderm, and it is only at the onset of gastrulation that the endodermal domain

will expand (Fig. 1C). This raises the possibility that the endoderm could be specified by the interaction between mesoderm and ectoderm. Together, this in situ hybridization screen shows that there is an approximately 4 h delay between the onset of the mesodermal marker gene expression at 6–8 hpf and the start of the endodermal marker gene expression at 10–12 hpf (Fig. 1D).

### β-catenin signaling promotes ectodermal and restricts mesodermal identity at the future oral pole

The delay in the onset of the endodermal marker gene expression in comparison to the mesodermal markers suggested a difference in their regulation and prompted us to investigate Wnt/β-catenin and MAP kinase signaling pathways, which were previously implicated in the development of the mesoderm in *Nematostella*. Some studies[14,19,20] suggested that β-catenin signaling promotes the specification of mesoderm, however, others presented experimental evidence showing that it antagonizes mesoderm formation[18,21]. Inactivation of MAPK signaling also perturbed mesoderm development and gastrulation movements[19], however, the role of MAPK in relation to Wnt signaling in germ layer formation was not addressed in detail.

To clarify the role of the Wnt/β-catenin pathway in the process of mesoderm and endoderm formation, we analyzed the expression of earliest mesodermal genes by in situ hybridization upon down- or up-regulation of the Wnt/β-catenin pathway (Fig. 2A, B). We analyzed the effects of these treatments at 8 hpf, when only ectoderm and mesoderm are specified, and at 20 hpf, early gastrula, when all three germ layers are present and mesoderm starts to invaginate during normal development. While untreated embryos display localized expression of *fgfa1, tbx19-like, duxABC* and *pitx1-like* in the future mesoderm at 8 hpf, knockdown of β-catenin by injection of a translation-inhibiting morpholino results in an ectopic expression of these mesodermal genes throughout the embryo, while ectodermal markers such as *koza1-like* and *APC* are downregulated (Fig. 2C). Conversely, ectopic upregulation of the Wnt/β-catenin pathway by treatment with the GSK3β inhibitor Azakenpaullone (AZ) abolishes mesodermal marker expression and leads to ubiquitous expression of ectodermal markers compared to control (Fig. 2C). Hence, in line with the earlier reports[18,21], we conclude that at this early stage β-catenin represses the mesodermal program and promotes the ectodermal program, restricting the mesoderm to the domain at the animal pole, where nuclear β-catenin is not expressed. The anti-mesodermal effect of β-catenin persists to later stages. At 20 hpf, mesodermal genes extend throughout the embryo upon β-catenin knockdown and disappear after upregulation of the β-catenin pathway by AZ treatment (Fig. 2D). We also observe that the expression of ectodermal markers (*foxQ2a and six3/6*) is abolished when the Wnt signaling is either down-regulated or upregulated. This result suggests that both mesodermal and endodermal programs have the ability to repress the ectodermal fate (Fig. 2D). The striking difference between the effects of β-catenin pathway manipulation at 8 hpf and 20 hpf is its effect on the emerging endoderm. Expression of the endodermal markers (*foxA* and *brachyury)* is abolished upon β-catenin knockdown and extends throughout the whole embryo in azakenpaullone-treated embryos, indicating that endodermal identity depends on β-catenin signaling. (Fig. 2E). Together, we confirm that β-catenin signaling suppresses mesodermal fate as well as promotes ectodermal and, at later stages, endodermal fate[13,21,22].

### MAPK signaling is essential for mesoderm induction and invagination

Based on previous studies, the MAP kinase pathway emerges as a potential candidate in the process of mesoderm formation[23,24]. To elucidate the role of the MAP kinase pathway (Fig. 3A), we identified the areas of active MAPK signaling during mesoderm formation by immunolabelling the activated form of ERK (pERK) and by in situ

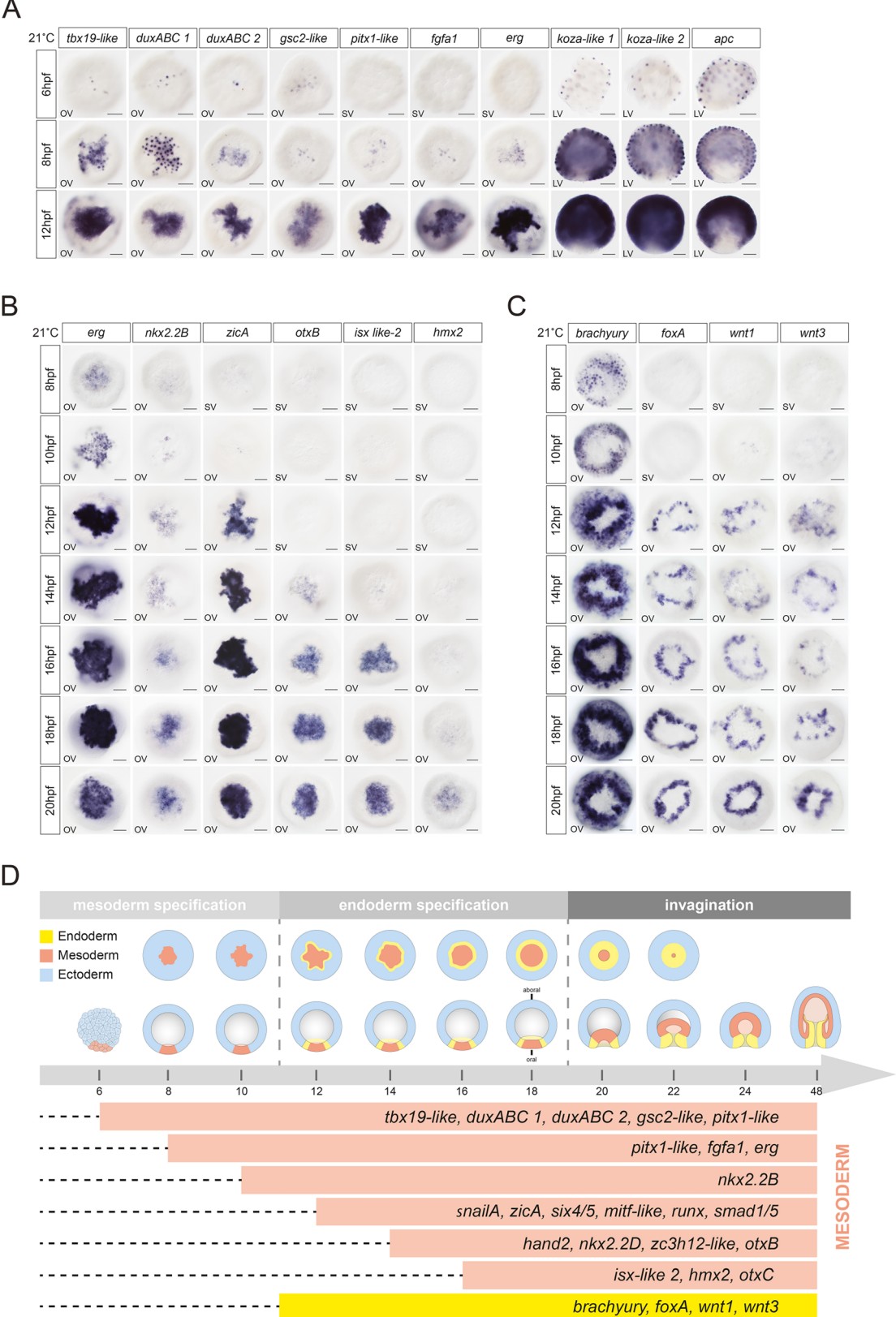

**Fig. 1 | Expression profiles of ectodermal, mesodermal, and endodermal genes at early embryonic stages in *Nematostella*. A** Spatio-temporal expression pattern of mesodermal and ectodermal markers at 6, 8, and 12 hpf revealed by in situ hybridization. Scale bar 50 μm. OV oral view, SV surface view. **B** Spatio-temporal expression patterns of mesodermal TFs at 8, 10, 12, 14, 14, 16, 18, and 20 hpf. Scale bar 50 μm. LV lateral view. OV oral view, SV surface view. **C** Spatial and temporal expression profiles of the endodermal TFs during early development. Scale bar 50 μm. OV oral view, SV surface view. **D** Summary of temporal succession of mesodermal and endodermal gene expression profiles during early development of *Nematostella*. All experiments were replicated three times with similar results.

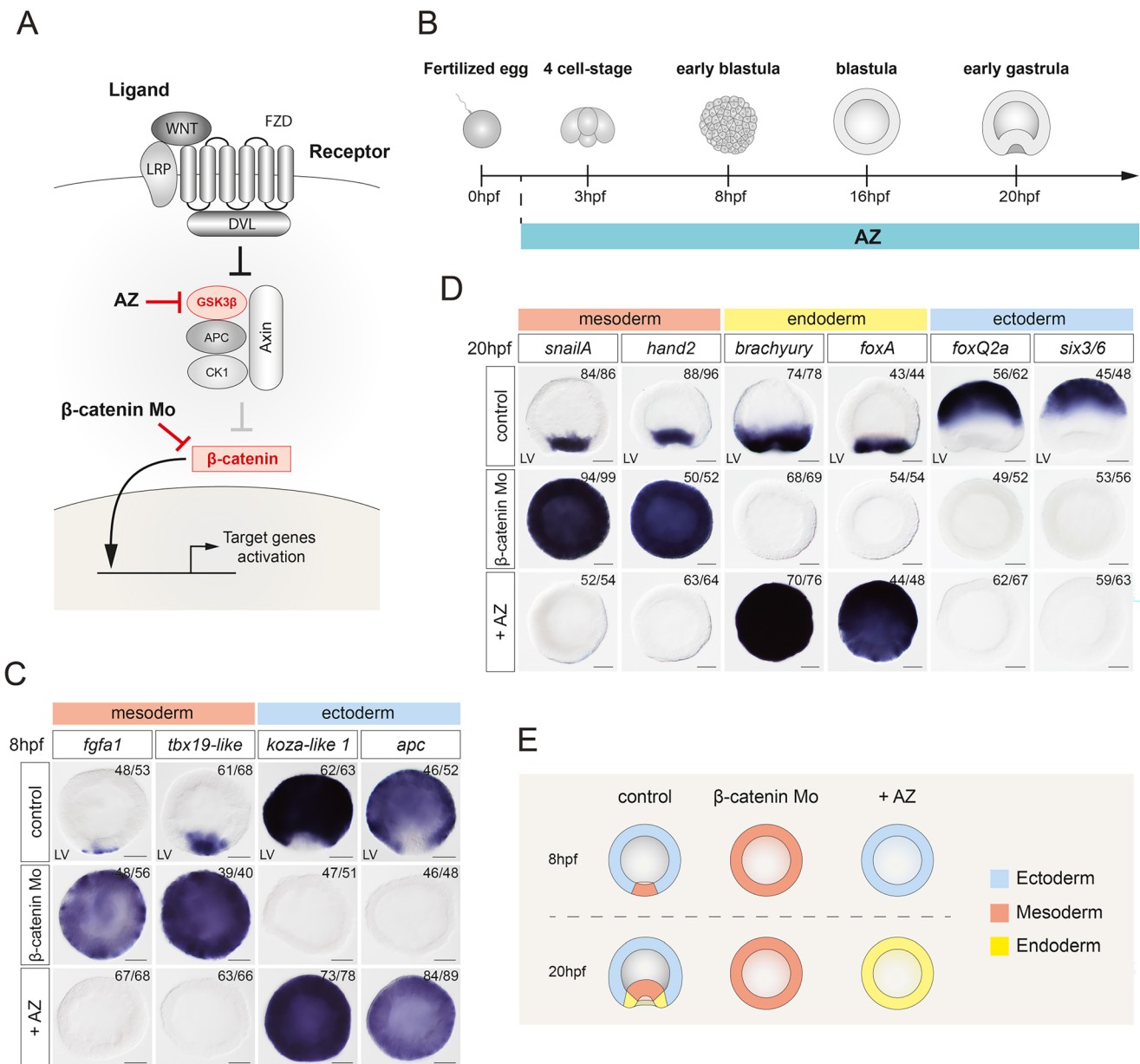

**Fig. 2 | β-catenin signaling restricts mesoderm fate to the oral pole. A** Scheme of the Wnt/β-catenin signaling pathway and the effect of the GSK3β inhibitor Aza-kenpaullone (AZ) and of β-catenin morpholino. **B** Scheme of treatment with AZ during the embryonic development. **C** In 8 hpf embryos, mesodermal markers *fgfa1* and *tbx19-like* become ubiquitously expressed upon β-catenin knockdown and are abolished upon AZ-mediated β-catenin stabilization. Ectodermal markers *koza-like1*

and *apc* demonstrate an opposite effect. Scale bar 50 μm. LV lateral view. **D** Expression of mesodermal (*snailA*, *hand2*), endodermal (*brachyury*, *foxA*), and ectodermal (*foxQ2a*, *six3/6*) genes in AZ-treated embryos or β-catenin morpholino embryos at 20 hpf. Scale bar 50 μm. LV lateral view. **E** Scheme describing the fate map changes at 8 hpf and 20 hpf induced by up- and downregulation of β-catenin signaling. All treatments were replicated three times with similar results.

hybridization against the MAPK target gene *erg*. We were able to detect pERK from 6–8 hpf in the nuclei of the mesodermal cells, promptly followed by the expression of the MAPK target gene *erg* (Fig. 3B). Western blot analyses confirmed that the MEK inhibitor U0126 can inhibit MAPK signaling and abolish the phosphorylation of ERK within 30 min of treatment (Fig. 3C, D). Embryos treated with U0126 upon fertilization showed a loss of expression of mesodermal markers *erg*, *tbx19-like*, and *fgfa1* (Fig. 3E). This observation at early blastula suggests that the MAP kinase pathway is required for mesoderm specification.

Prolonged treatment with U0126 until the onset of gastrulation (20 hpf) resulted in the abolishment of most of the mesodermal genes, although *erg*, *six4/5*, and *zicA* maintained their expression in a smaller oral domain (Fig. 4A and Supplementary Fig. 4A). By comparison, the

expression of endodermal markers, such as *foxA* and *wnt3* is still present in U0126-treated embryos. Moreover, we observe ectopic expression of *brachyury* and *foxA* in the mesodermal territory, suggesting that these genes are normally suppressed in the mesoderm by MAPK signaling (Fig. 4A, see also ref. 23). Similar results were obtained after downregulation of the *erg* expression, one of the key transcription factors downstream of the MAPK signaling[24] (Supplementary Fig. 4B).

Although changes in gene expression show that inhibition of the MAPK pathway by U0126 treatment affects the specification of mesodermal identity at the molecular level, the mesoderm nevertheless becomes morphologically distinct at the time of gastrulation. However, gastrulation is arrested shortly after the onset (Fig. 4B, D). During normal gastrulation, mesodermal cells become loosely packed,

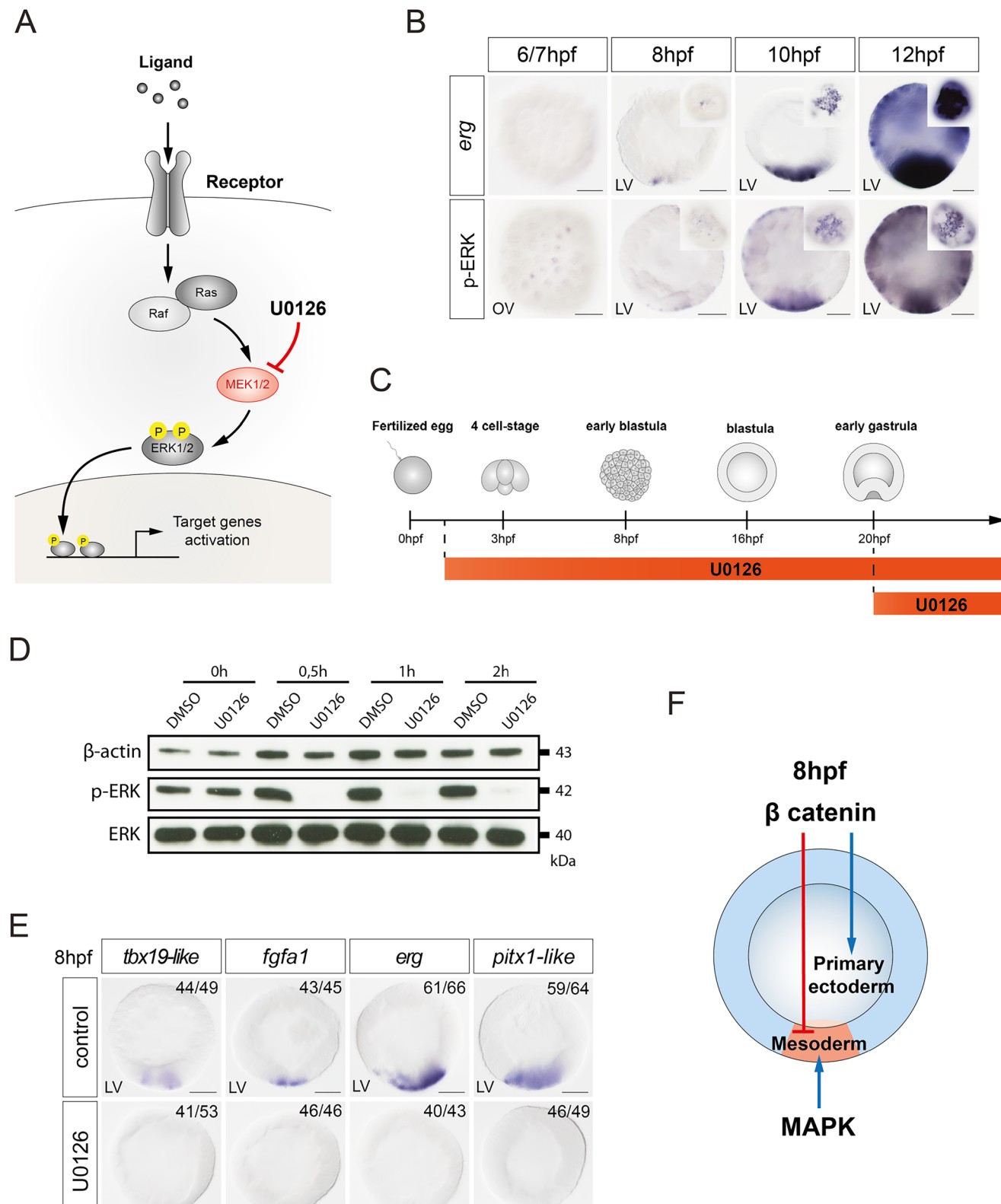

**Fig. 3 | MAPK signaling is essential for mesoderm specification. A** Scheme of the RTK/MAPK signaling pathway showing the role of the MEK inhibitor U0126. **B** Spatial and temporal expression profile of *erg* revealed by in situ hybridization in comparison to phospho-ERK immunostaining at 6, 8, 10, 12 hpf. Scale bar 50 μm. LV lateral view. OV oral view. **C** Scheme of treatments with U0126. **D** Western blot analysis of lysates of DMSO- or U0126-treated 20 hpf embryos shows that U0126 efficiently blocks ERK phosphorylation after a 30 min treatment. **E** Mesodermal gene expression is suppressed by the U0126-mediated inhibition of MAPK signaling at 8 hpf. Scale bar 50 μm. LV lateral view. All treatments were replicated three times with similar results. **F** Schematic summary of the role of β-catenin and MAPK signaling in the specification of mesoderm. Primary ectoderm – ectoderm prior to patterning by a Wnt signaling gradient.

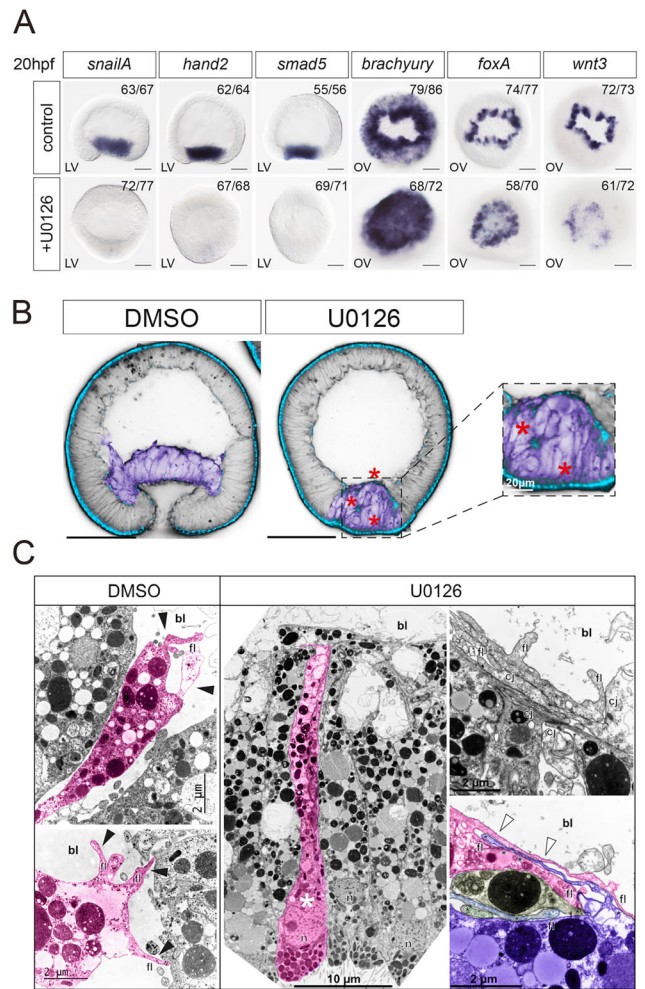

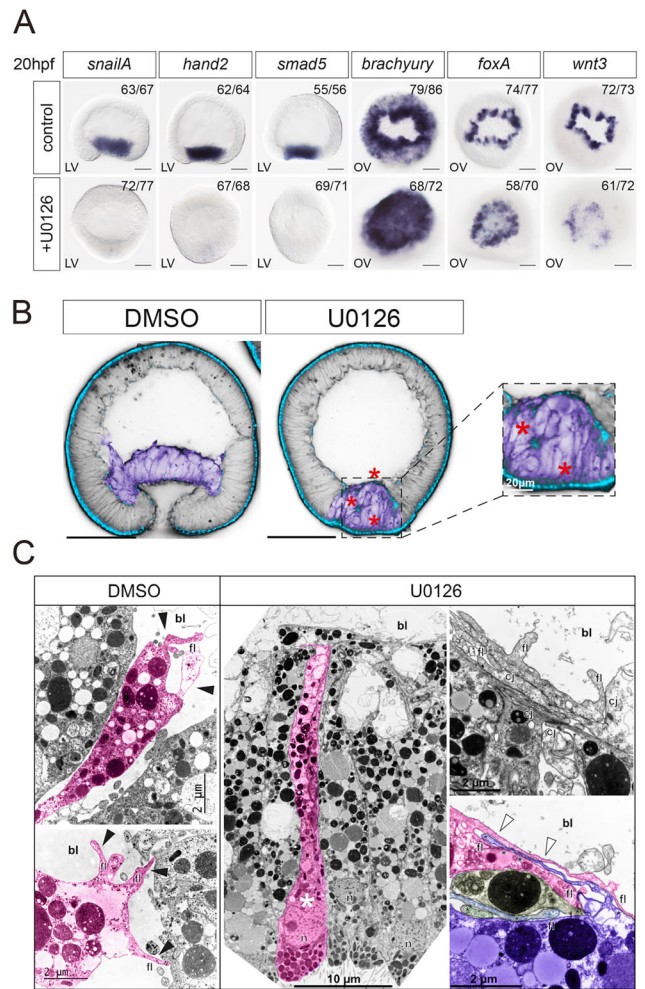

**Fig. 4 | MAP kinase signaling is essential for mesoderm invagination. A** At 20 hpf, MAPK signaling inhibition with U0126 abolished expression of mesodermal markers and expanded expression of endodermal markers (*brachyury*, *foxA*) into the mesodermal domain. Scale bar 50 μm. LV lateral view, OV oral view. **B** Ectodermal cadherin cdh3 (blue) does not disappear from the mesodermal cells upon MAPK signaling inhibition (red asterisk). Cell outlines are shown by fibrillar actin staining (black). Mesodermal cells are highlighted in purple. Scale bar 100 μm. All treatments and staining experiments were replicated three times with similar results. **C** TEM comparison of the U0126-treated and control DMSO-treated embryos at the early gastrula stage shows that in U0126, mesodermal cells do not become bottle-shaped (white asterisk), and their filopodia appear to stick together and form numerous contacts rather than extend into the blastocoel (white arrowheads), unlike filopodia from embryos treated with DMSO (black arrowheads). bl blastocoel, fl filopodia, cj cell junction, n nucleus. **D** SEM images of the U0126 and DMSO-treated embryos at the mid-gastrula stage. Note the lack of mesoderm invagination into the blastocoel and the absence of extending filopodia in U0126-treated embryos (white asterisk).

acquire bottle-cell morphology, and form filopodia reaching to the basal surfaces of the ectodermal cells as the mesoderm and ectoderm extend their contact in a zipping motion[25,26]. Ultrastructural analyses of U0126-treated embryos show that the mesodermal cells start showing signs of the incomplete EMT (e.g., onset of the cell shape change, nuclear migration) but fail to disassemble their basal contacts and extend filopodia to the basal surfaces of the ectodermal cells (Fig. 4B, D).This suggests that cell-cell adhesion is affected by the inhibition of the MAPK pathway. Normal gastrulation in *Nematostella* is associated with a switch of Cadherin3 to Cadherin1 in the mesoderm[27]. Embryos with MAPK pathway inhibited through treatment with U0126 or knockdown of *ERG* maintain Cadherin3 expression in the mesoderm, which may explain the retention of basal contacts between the mesodermal cells, the lack of the filopodia extension and of the migration of the mesodermal cells along the blastocoel roof (Fig. 4B and Supplementary Fig. 4C). Thus, MAPK signaling is essential to the specification of the mesoderm and subsequent morphogenetic processes required for gastrulation.

## The kinase Mos2 is a potential activator of the MAPK signaling in the future mesoderm

Given the critical role of the MAPK signaling pathway in the formation of the mesodermal identity in *Nematostella*, it is important to elucidate the mechanism that induces its activation. In the sea urchin, KSR3, a serine/threonine kinase, has been recently identified as responsible for the MAPK signaling activation in the mesoderm independently of the ligand/receptor interaction[28]. To identify kinases with similar functions in *Nematostella*, we next interrogated the single-cell transcriptomes of early developmental stages. Interestingly, the genome of *Nematostella* contains 4 Mos genes. Mos kinases are a family of serine/threonine kinases that are highly conserved among Eumetazoa and are able to activate MAPK signaling[29]. Mos kinases are typically expressed in the oocytes of many organisms, where they are involved in postmeiotic arrest[30]. While only a single gene coding for *mos* is found in bilaterians, 2 *mos* have been identified in the cnidarian *Clytia*[29]. Chmos1 and Chmos2 kinases can activate MAPK signaling when they are overexpressed in *Xenopus*[29], suggesting a conserved molecular function.

Based on single-cell RNA seq, *Nematostella mos2* appears to be expressed in the mesodermal cells. We confirmed the single cell profile by in situ hybridization and found that *mos2* starts from 8hpf through gastrulation in the mesoderm (Supplementary Fig. 3B). To investigate the potential role of Mos2 in the activation of the MAPK signaling, we assayed for mesodermal gene expression following the injection of *mos2* mRNA into fertilized eggs (Supplementary Fig. 3C, D). Indeed, *mos2* overexpression induces ectopic expression of the mesodermal gene *erg* at 10 hpf and *snailA* at 48 hpf throughout the embryo (Supplementary Fig. 3C,D). At 72 hpf *mos2*-injected embryos show partial mesoderm extension at 250 ng/μL, comparable to low-dose β-catenin morpholino injection, and full mesoderm extension throughout the whole embryo at 500 ng/μL (Supplementary Fig. 3E). These findings support a potential role for Mos2 in the upstream activation of the MAPK signaling responsible for mesoderm initiation (Supplementary Fig. 3F).

## MAPK and β-catenin signaling suppress the default aboral ectodermal identity of the embryo

Previous analyses showed that β-catenin-mediated endodermal identity overrides the originally ubiquitous aboral ectodermal fate[13]. However, suppression of β-catenin signaling results not in the expansion of the aboral ectoderm but in a ubiquitous expression of MAPK-dependent mesodermal markers[21] (this study). To identify the epistatic relationship between β-catenin and MAPK signaling, we injected zygotes with β-catenin morpholino, which normally should activate the mesodermal program throughout the entire embryo and downregulate endodermal genes, but in parallel we inhibited the MAP kinase pathway using U0126 and analyzed the expression of marker genes for the endoderm, the mesoderm and the aboral ectoderm (Fig. 5A). In this condition of simultaneous β-catenin knockdown and MAPK inhibition, neither mesodermal nor endodermal markers were expressed. Instead, zygotic markers of the aboral ectoderm *six3/6* and *foxQ2a* were expressed ubiquitously (Fig. 5A, B). This is striking since earlier studies showed that aboral ectoderm markers are abolished in the β-catenin morphants[21,22]. Moreover, ubiquitous expression of *six3/6* upon simultaneous inhibition of MAPK and β-catenin signaling appears similar to what happens when aboral cells, isolated by bisection at the eight-cell stage, grow in the absence of signals from the oral domain[21]. Our new results suggest that (i) MAPK-dependent suppression of *six3/6* and *foxQ2a* aborally is caused by the expansion of the mesoderm in the β-catenin morphants; (ii) support earlier results suggesting that aboral ectoderm represents the default state of the embryo in the absence of MAPK and β-catenin signaling; and iii) indicate that MAPK signaling activity is what activates mesodermal genes in the β-catenin morphants (Fig. 5B).

## Endoderm segregation requires Notch signaling

In *Nematostella*, the endoderm emerges between the mesoderm and ectoderm around 12–14 hpf. Since the Delta/Notch pathway is known to play a role in the segregation of mesoderm and endoderm in other organisms[31], we investigated its function in endoderm specification in *Nematostella*. First, we conducted an expression analysis of the receptor *notch* and its potential ligands at multiple time points from egg to 26 hpf using in situ hybridization. While only one gene coding for the Notch receptor has been previously found in *Nematostella*[32], four ligands of this pathway, *delta*, *delta-like*, *jagged1-like*, and *jagged1B-like*, can be identified in the genome. In situ hybridization shows that *notch* mRNA is maternally deposited in the egg and persists in the embryo. It is initially expressed ubiquitously, but then its expression disappears from the mesoderm around 10 hpf (Fig. 6A and Supplementary Fig. 5A). After gastrulation, Notch expression is downregulated in the midbody ectoderm[32] (Supplementary Fig. 5). mRNAs coding for *delta* and *delta-like* are zygotic, but they are also expressed before the onset of gastrulation, at 10 and 14 hpf, respectively. Both are primarily found in the mesoderm, with *delta-like* mRNA

transcripts visualized from 10 hpf (Fig. 6A, Supplementary Fig. 5). In addition, *delta* (but not *delta-like*) is also detected at the aboral side in a salt and pepper pattern after 12 hpf, likely reflecting its role in early neurogenesis in this domain[32,33]. The early mesodermal expression of *delta* shows a peak between 14–16 hpf before its expression gradually decreases towards gastrulation (Fig. 6A and Supplementary Fig. 5A). The third Notch ligand gene, *jagged1-like*, exhibits a later, and rather weak expression starting at 18 hpf in the mesoderm (Supplementary Fig. 5A). While *delta* and *jagged1-like* expression decreases in the mesoderm after 18 hpf, *delta-like* expression starts at 16 hpf and continues to be abundantly detected in the mesoderm after gastrulation (Supplementary Fig. 5A). Since the last Notch ligand gene *jagged1B-like*, based on single cell RNA sequencing data, is expressed at very low level uniformly across all cell clusters, we did not investigate its expression at early stages. Thus, Notch ligand genes *delta, delta-like*, and *jagged1-like* are mostly expressed in the mesoderm, whereas *notch* is expressed in the ectoderm. Double in situ hybridization shows that the expression of *delta* in the mesoderm and *notch* in the whole ectoderm of the embryo are precisely complementary, without any overlap (Fig. 6B). To assess, whether endoderm is induced on the mesodermal or the ectodermal side of the *notch-delta* boundary, we analyzed the expression of *notch* together with the endodermal marker *foxA*. We detect *foxA* expression in the *notch*-expressing ectodermal border cells (Fig. 6C), which are in direct contact with the *delta*-expressing mesodermal cells (Fig. 6B). These results suggest that endoderm is ectoderm-derived, and that Notch signaling may have a role in endoderm induction.

The activated Notch pathway is marked by the translocation of the Notch intracellular domain (NICD) to the nucleus (Fig. 6D). To check, which cells and domains in the embryo show an activated Notch pathway, we carried out immunohistochemistry against the NICD in early embryos. Strikingly, we find two domains in early gastrula stage embryos: the first is an uninterrupted ring at the blastopore corresponding to the endoderm/mesoderm *Notch* expression border at late blastula stage (Fig. 6D); the second, appearing several hours later, is in small cell patches in the aboral half, consistent with the *delta* expression and the potential role of Notch signaling in specifying neuronal precursor cells (Fig. 6D[32–34]). Complementary expression of Delta ligand and Notch in the mesoderm and ectoderm, respectively, suggested their dependence on β-catenin. Indeed, upon morpholino-mediated knockdown of *β-catenin, delta* and *delta-like* expression became ubiquitous, while *notch* expression was lost (Supplementary Fig. 5B). Conversely, overactivation of β-catenin signaling by AZ treatment results in the loss of *delta* and *delta-like* and the extension of *notch* expression throughout the whole embryo. Thus, *delta* and *delta-like*, like other mesodermal genes, are repressed, while *notch*, like other early ectoderm genes, is activated by the β-catenin signaling (Fig. 6E and Supplementary Fig. 5B). By contrast, a knockdown of the mesodermally expressed MAPK-dependent transcription factor *erg*, did not lead to significant changes in the expression of the *delta* gene. However, *erg* knockdown induces ectopic expression of *notch* in the mesoderm, suggesting that although ERG does not contribute significantly to the regulation of *delta* expression it may still be involved in the repression of *notch* in the mesoderm (Supplementary Fig. 5B).

## Endoderm is induced by Delta-Notch signaling between the ectoderm and the mesoderm

The formation of the endoderm on the *notch* side of the *delta-notch* expression boundary raises the possibility that the activation of the Notch pathway by Delta induces endoderm formation. To test this, we inhibited the Notch pathway with the γ-secretase inhibitor LY411575, which prevents the release of the intracellular domain of the Notch receptor (NICD) (Fig. 7A). Upon treatment with increasing concentrations of the Notch signaling inhibitor, the expression of endodermal genes *brachyury, foxA, wnt1* and *wnt3* at 20 hpf was repressed in a

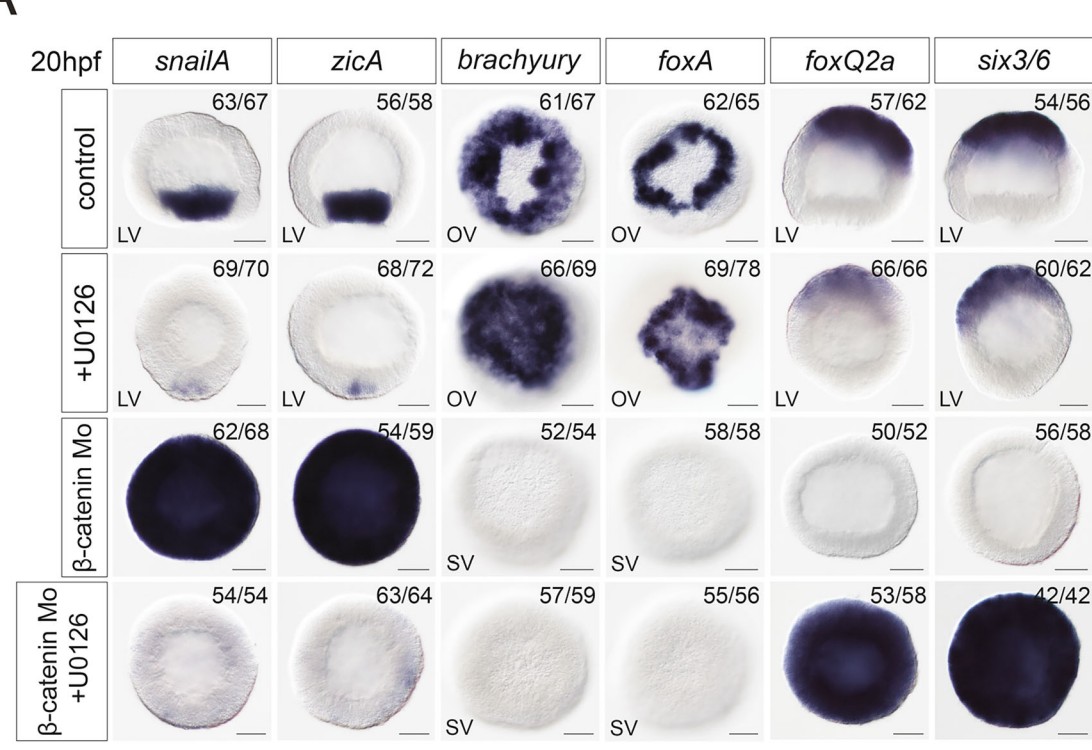

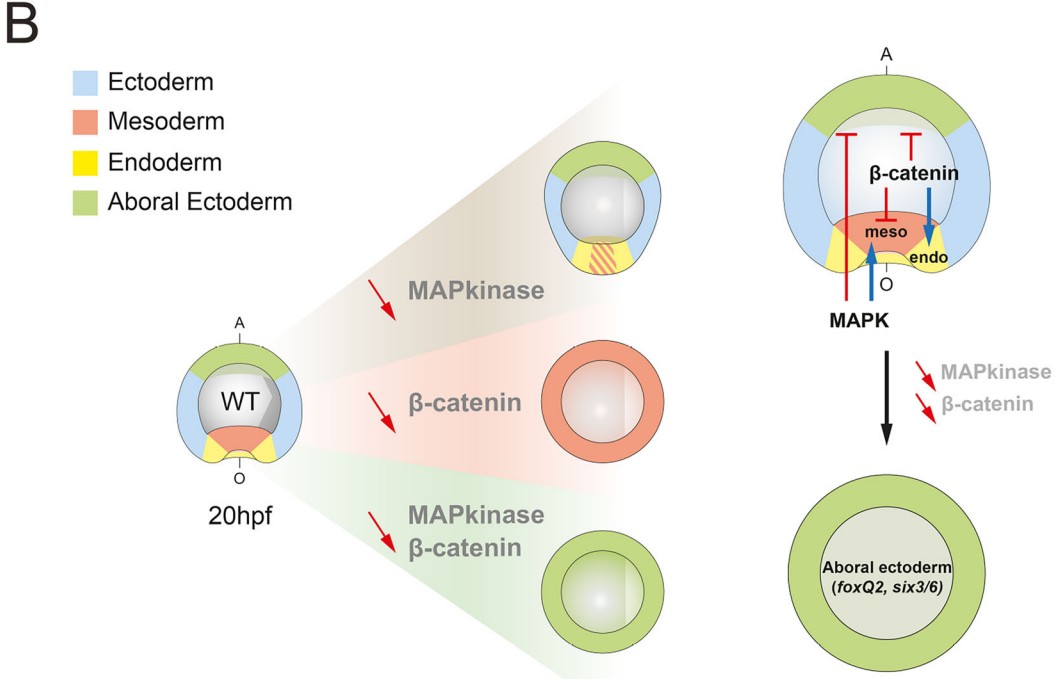

**Fig. 5 | MAP kinase signaling acts in concert with β-catenin signaling to inhibit aboral ectoderm.** **A** At 20 hpf, MAPK and β-catenin signaling inhibition abolishes mesodermal and endodermal markers but expands ubiquitously the expression of aboral ectodermal markers (*foxQ2a, six3/6*). Scale bar 50 µm. LV lateral view, OV oral view, SV surface view. All treatments were replicated three times with similar results. **B** Schematic summary of single or combined knockdown/inhibition of β-catenin and MAPK signaling on aboral ectoderm marker genes (*foxQ2a, six3/6*).

concentration-dependent manner (Fig. 7B). By contrast, Notch signaling inhibition had no effect on the expression of the mesodermal genes, such as *snailA* or *zicA* (Fig. 7B). In line with the previous observations, the disappearance of the *wnt-* and *brachyury-* expressing endodermal

fate did not affect gastrulation but led to defects in the Wnt-dependent patterning of the ectoderm and affected pharynx formation: we observed the shift of the midbody marker *Wnt2* expression towards the oral end and the expansion of the aboral ectoderm markers *foxQ2a* and

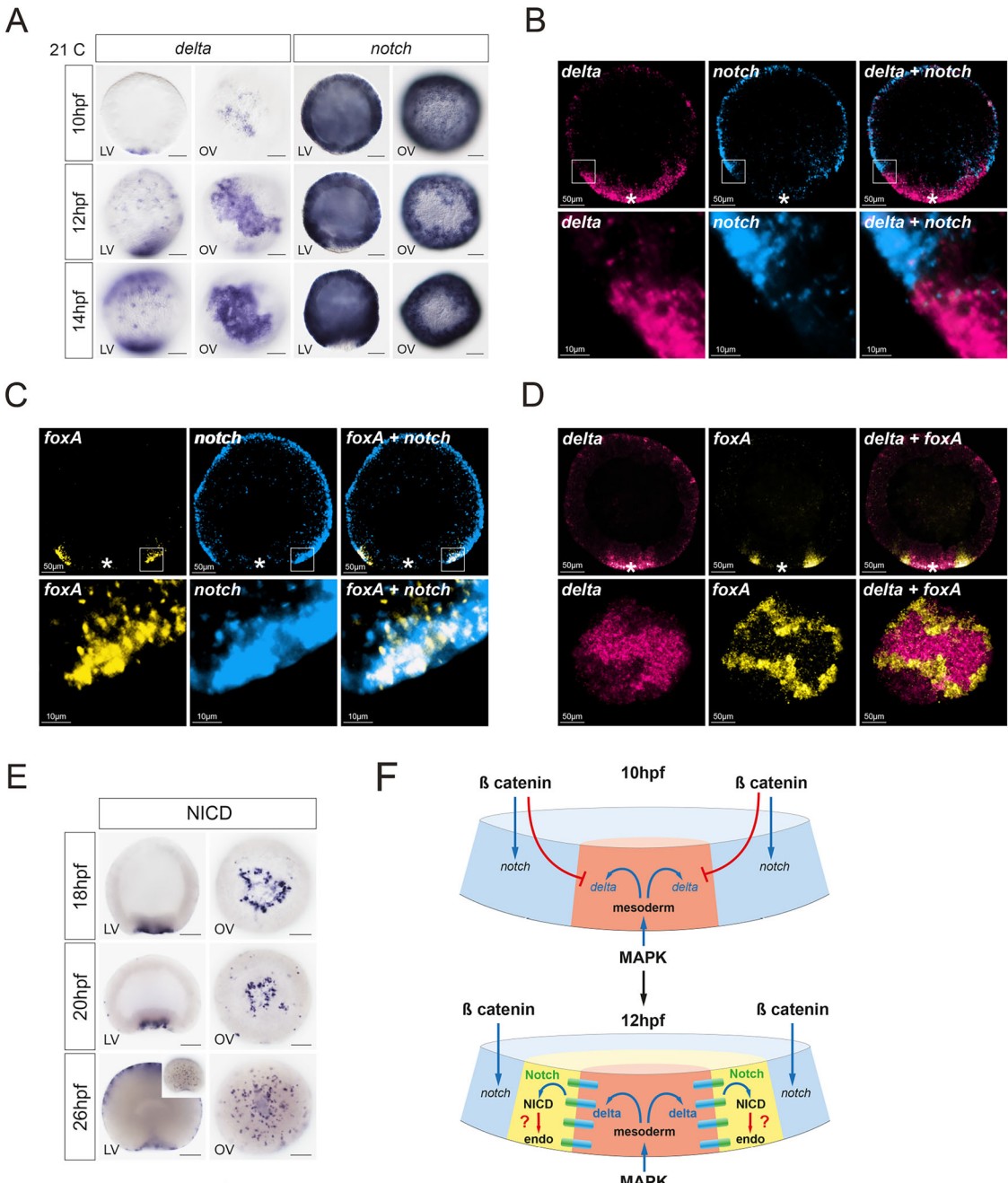

**Fig. 6 | Delta and Notch expression demarcate the mesoderm/ectoderm border. A** Expression profile analysis of the ligand *delta* and receptor *notch* from 10 hpf to 14 hpf by in situ hybridization. While *delta* (and *delta-like)* are expressed mostly in the mesoderm, *notch* is detected in the ectoderm. Scale bar 50 µm. LV Lateral view, OV oral view. **B** Fluorescent double in situ hybridisation with probes against *delta* (magenta) and *notch* (cyan). *delta* (mesoderm (white asterisk)) and *notch* (ectoderm) are expressed in a complementary abutting pattern. **C** Fluorescent double in situ against *foxA* (yellow) and *notch* (cyan). *foxA* is expressed in the *notch*-expressing ectodermal border cells, which are in contact with the mesoderm (white asterisk). **D** Fluorescent double in situ against *delta* (magenta) and *foxA* (yellow). *foxA* (future endoderm) and *delta* (mesoderm (white asterisk)) are expressed in non-overlapping spatial domains. **E** Activated Notch pathway is revealed with NICD immunostaining at 18, 20, and 26 hpf. Scale bar 50 µm. LV lateral view, OV oral view. **F** Schematic summary of the regulation of *delta* and *notch* expression by β-catenin and MAPK signaling followed by endodermal genes activation by the Notch pathway. All stainings were replicated three times with similar results.

*six3/6* at the gastrula stage, and suppression of the endodermal pharynx development at the 48 hpf planula stage (Fig. 7B,C, Supplementary Fig. 6[13,21,22,35,36]). Thus, we conclude that Notch signaling is necessary for endodermal fate induction in the ectodermal cells located at the border to the mesoderm.

To test whether Notch signaling is also sufficient to induce endoderm, we ectopically activated Notch signaling in the ectoderm by mosaically overexpressing either *NICD-mCherry* or, as a negative control, just *mCherry* under control of the ubiquitously active *TBP*

promoter using the established transgenesis protocol[37,38]. The visualization of mCherry expression showed a uniform distribution throughout the cells, whereas NICD-mCherry clearly suggested nuclear translocation (Fig. 7D). Strikingly, injection of *TBP::NICD-mCherry* plasmid resulted in ectopic expression of *foxA* and *brachyury* in 60% of cases, whereas no ectopic endodermal marker expression was observed with the *TBP::mCherry* plasmid (Fig. 7E). The expression of *snailA* was not affected, indicating that NICD specifically activates endodermal but not mesodermal genes (Fig. 7E). Taken together, these

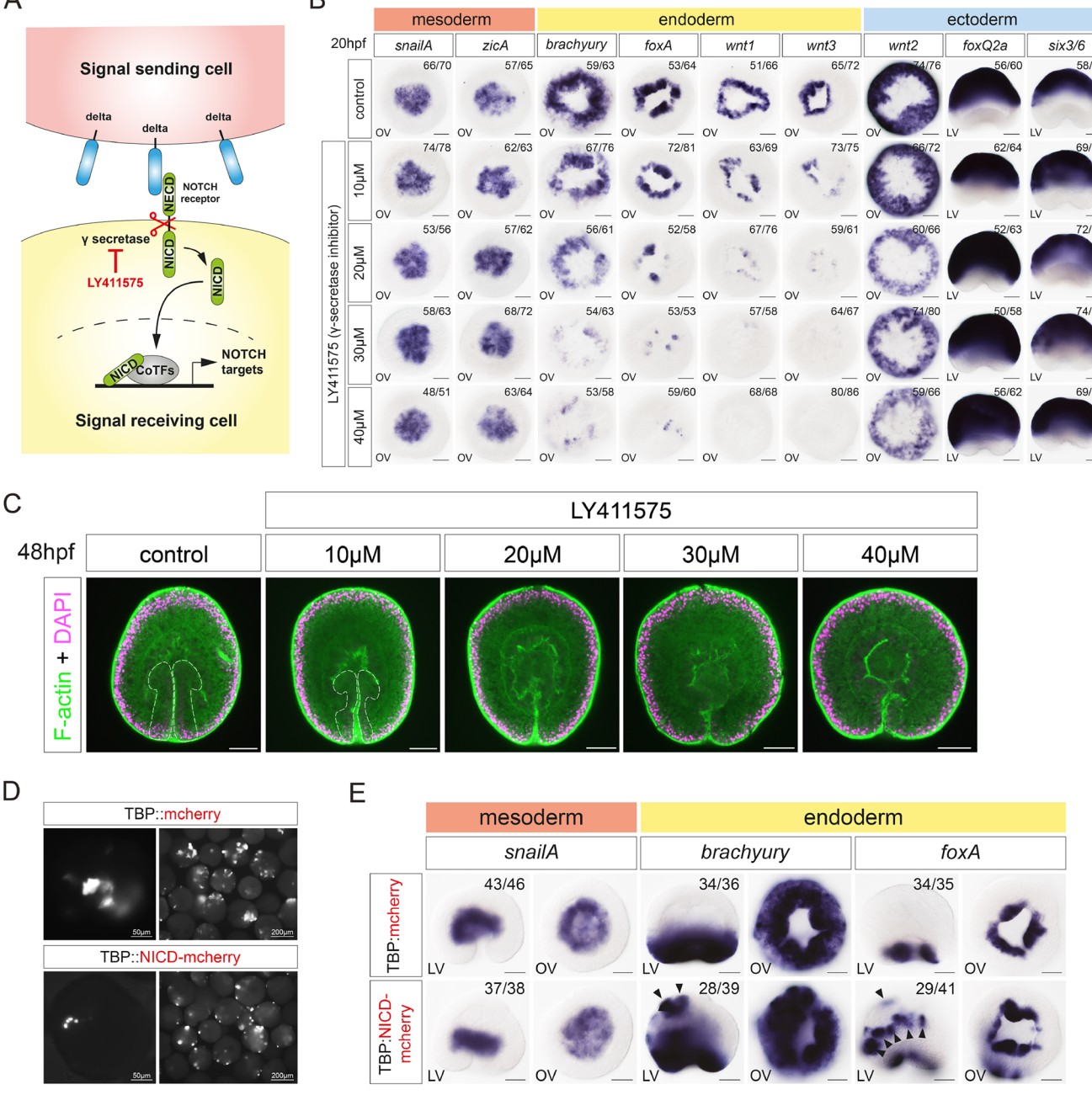

**Fig. 7 | Notch signaling pathway plays a crucial role in endodermal tissue formation by activating endodermal genes. A** Scheme of Notch signaling and the target of the inhibitor LY411575. **B** Visualization of mesodermal (*snailA, zicA*), endodermal (*brachyury, foxA, wnt1, wnt3*), and ectodermal (*wnt2, foxQ2a, six3/6*) gene expression at 20 hpf on embryos treated with LY411575 using in situ hybridization; Expression of endodermal genes is downregulated dose-dependently upon inhibition of the Notch signaling. Scale bar 50 μm. LV lateral view, OV oral view. **C** Morphological analysis on LY411575-treated embryos stained with phalloidin (green) and DAPI (magenta). LY411575 treatment induces the loss of the endoderm. Scale bar 50 μm. **D** mCherry expression in *TBP::mCherry* and *TBP::NICD-mCherry* injected embryos at 24 hpf. **E** Analysis of mesodermal (*snailA*) and endodermal (*brachyury, foxA*) gene expression in *TBP::mCherry* and *TBP::NICD-mCherry* injected embryos at 20 hpf. Black arrowheads show ectopic expression of endodermal markers when NICD is overexpressed. Scale bar 50 μm. LV lateral view, OV oral view. The experiments were replicated three times with similar results.

experiments show that Notch signaling is both necessary and sufficient to induce endodermal genes in the Notch-expressing ectodermal domain, consistent with the idea that the abutting expression of mesodermal *delta* and ectodermal *notch* at 10–14 hpf leads to the induction of endoderm.

## Discussion

In this study, we investigated how three signaling pathways – β-catenin, MAPK, and Notch – interact and function during the specification of the three germ layer identities in *Nematostella*. We show that in *Nematostella*, mesoderm is the first embryonic territory to be specified in the 6 hpf blastula stage embryo, which has ubiquitous aboral ectoderm identity as a default state. While the activation of the mesoderm specification program in a very precisely defined β-catenin-negative domain appears to rely on yet unknown maternal components[18], the role of β-catenin signaling in the early blastula is to prevent mesodermal gene expression outside of the mesodermal domain (ref. 21; this study) and, as we show here, to promote expression of the early

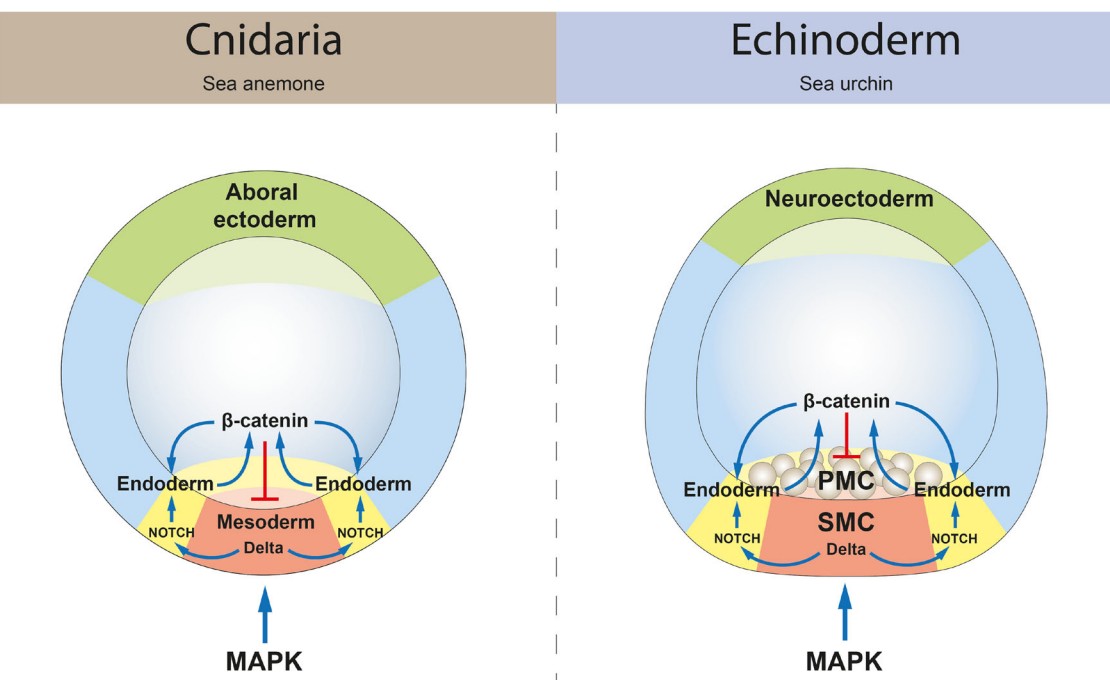

**Fig. 8 | Comparison of interactions of signaling pathways in the specification of the sea anemone *Nematostella* and sea urchins.** Figure of an echinoderm adapted from ref. 72.

ectodermal markers outside the mesoderm. Using pharmacological treatments and gene knockdown, we demonstrate that MAPK signaling is responsible not only for the activation of the mesodermal marker expression but also for the repression of the zygotically expressed aboral ectoderm markers such as *six3/6* and *foxQ2a*. Finally, we show that endodermal fate is induced in a single row of cells at the interface between the Notch-expressing ectoderm and the Delta-expressing mesoderm. Moreover, Notch signaling activation is necessary and sufficient for the initiation of the endodermal gene expression program in any ectodermal cell. We consider the latter two findings highly significant and will discuss them below.

### MEK/MAP kinase signaling, an ancestral and conserved pathway to induce mesoderm

We showed that activation of the MAP kinase signaling is essential for the specification of the mesoderm in the cnidarian *Nematostella*. Its inhibition with a selective inhibitor of MEK disrupts early mesodermal gene expression and causes gastrulation defects. We see a similar situation in some Bilateria, especially in ambulacrarian deuterostomes (Hemichordata and Echinodermata), whose germ layer fate maps – mesodermal cells at the vegetal pole surrounded by a concentric endodermal domain, and an ectoderm making up the rest of the embryo beyond the endoderm – are strikingly similar to the germ layer fate map of *Nematostella* (Fig. 8), except that in *Nematostella* meso- and endoderm form at the animal rather than at the vegetal pole[39–42], MAPK signaling was shown to be important for the formation of the primary and of a subset of the secondary mesenchymal cells in sea urchin *Paracentrotus*[43], and for the mesoderm specification in the hemichordates *Saccoglossus* and *Ptychodera*[44–46]. The situation in non-vertebrate chordates is somewhat different. FGF-dependent MAPK signaling is responsible for the mesoderm induction by the future endodermal cells in the urochordate *Ciona*[47,48], while in the cephalochordate *Branchiostoma* only anterior somitic mesoderm appears to form in a MAPK-dependent manner[49]. In vertebrates, Nodal and TGFß signaling are responsible for mesoderm induction, however, FGF-mediated MAPK signaling (together with Wnt, BMP and Hedgehog signaling pathways) is involved in the mesodermal patterning[50,51]. In

protostomes, the role of FGF-mediated MAPK signaling has been investigated in three lophophorate spiralian species: the brachiopods *Terebratalia* and *Novocrania* and in a related phoronid *Phoronopsis*. In the two brachiopods, pharmacological suppression of the FGF receptor resulted in embryos lacking coelomic mesoderm, and muscle tissue was also lost in the phoronid embryo[12]. In another spiralian, the annelid *Alitta*, MAPK inhibition led to defects in the mesodermal patterning and morphogenesis, however, it did not abolish the mesodermal fate[52]. MAPK signaling is well-known for its role in the specification of the 3D macromere in gastropods, such as the snails *Crepidula* and *Ilyanassa*[53,54]. However, the daughter cell of the 3D, the mesentoblast 4d, appears not to require MAPK signaling, and although heart and muscle are missing in *Ilyanassa* upon MEK inhibitor treatment, we cannot be certain that the role of MAPK signaling here is induction of mesoderm. Nevertheless, the role of MAPK signaling in mesoderm specification has been documented in ambulacrarian deuterostomes, in lophophorate spiralian protostomes, and now also in a cnidarian *Nematostella*. Thus, despite all the diversity, it appears plausible that the origin of MAPK-signaling as a mechanism of specification of mesoderm tissue identity may have predated the split of Cnidaria and Bilateria.

### β-catenin signaling does not promote the mesodermal fate in *Nematostella* as in deuterostomes

Our study demonstrates that the MAPK signaling is indispensable for mesoderm formation in *Nematostella*, while β-catenin signaling actively suppresses mesodermal fate specification. This result contradicts earlier studies proposing a pro-mesodermal role of the β-catenin signaling[19,20]. Previous support for this involvement of β-catenin signaling in the initiation of mesoderm derived from two observations, such as the presumed nuclear localization of β-catenin-GFP within presumptive endodermal and mesodermal territories after β-catenin GFP mRNA injection, and the loss of the mesodermal gene *snailA* upon downregulation of the Wnt receptor gene *frizzled 1*[19,20]. However, our data as well as recent studies challenged these conclusions. A knock-in line expressing endogenous β-catenin tagged with GFP revealed a nuclear β-catenin localization in the ectoderm, on the opposite side of

the gastrulation, excluding the future mesoderm[18]. Moreover, knockdowns of the orally expressed Frizzled 1, Frizzled 4, and Frizzled 10 as well as of the co-receptor LRP5/6 using shRNA, and morpholinos showed no effect on mesodermal gene expression while severely affecting endodermal and ectodermal marker genes[22]. Our results align with these newer findings, providing robust evidence that β-catenin signaling does not promote early mesoderm specification.

### Notch signaling induces endoderm specification in *Nematostella*

As mentioned above, in terms of mutual positions of the germ layers, the fate map of the early *Nematostella* embryo is strikingly similar to the fate maps of ambulacrarian deuterostomes. Despite the difference in the mechanism of the mesoderm specification (β-catenin-dependent endomesoderm specification followed by turning β-catenin signaling off in the mesoderm once it is specified and keeping β-catenin signaling on in the endoderm in Ambulacraria vs. mesoderm specification in a domain initially lacking nuclear β-catenin in *Nematostella*), the subsequent patterning of the endoderm and ectoderm by a gradient of Wnt/β-catenin signaling appears to use the same logic and activate orthologous downstream genes[13,18]. However, this raises an obvious question: when and how does *Nematostella* form the endoderm if mesoderm is specified in the β-catenin-negative domain of the early blastula, and the remaining maternal β-catenin-positive domain represents the ectoderm? Our data show that this occurs several hours later by induction.

One striking feature shared by ambulacrarian and cnidarian embryos is that pharmacological upregulation of β-catenin signaling leads to endodermal marker expression in the ectodermal cells, while mesodermal cells, once they are specified, become insensitive to changes in the β-catenin levels[13]. The best candidate signaling pathway for inducing a novel fate at the interface of two different populations of cells was the Delta/Notch pathway. In animal development, Delta/Notch signaling is a major facilitator of binary cell fate decisions using two distinct types of regulatory logic: lateral inhibition and lateral induction. During lateral inhibition, a cell within an equivalence group of equipotent cells expressing low levels of Notch and Delta is induced to acquire a certain fate, which makes it produce more Delta than its neighbors. This Delta-producing cell signals to the neighbors causing them to shut down Delta production and acquire an alternative fate[31]. This mechanism is often used in neural differentiation, likely including its previously reported role in the formation of the nervous system in *Nematostella*[32–34]. During lateral induction, scenarios may be different. In a field of equipotent cells expressing both Delta and Notch, lateral induction may result in concerted behavior of large groups of cells, as in vertebrate somitogenesis and, possibly, in arthropod segmentation[55]. However, at the interface between the Notch-expressing cell population and the Delta-expressing cell population, lateral induction leads to the emergence of the third cell state[56] – in *Nematostella*, this is the endoderm. In sea urchins, like in *Nematostella*, Notch is initially ubiquitous, and eventually suppressed in the mesodermal lineage, where Delta starts to be expressed; however, unlike in *Nematostella*, Delta/Notch signaling is responsible for the segregation of the specific mesodermal populations rather than for the induction of the endodermal fate[8,31,57,58]. In contrast, in sea stars, which lack the skeletogenic primary mesenchymal cells, Delta/Notch signaling is responsible for the formation of the endoderm/mesoderm boundary[10] despite a similar expression pattern of *Notch* and *Delta* as in sea urchins. Similarly, Delta signaling from mesoderm to endoderm and Notch-dependent suppression of the mesodermal gene expression in the endoderm have been hypothesized for cephalochordates[59]. Thus, whether Notch-dependent endoderm/mesoderm segregation represents a trait ancestral for Cnidaria and Bilateria remains difficult to judge. Delta/Notch signaling is an efficient boundary formation mechanism, which can be used differently in the process of endoderm-mesoderm segregation, even in the closely related animal clades.

However, mesodermal expression of *Delta* versus endo- and ectodermal expression of *notch* may indeed represent an ancestral condition.

### Comparison of germ layer identities in *Nematostella* and bilaterians

An earlier study has challenged long-held views about homologies of cnidarian and bilaterian germ layers and postulated that the inner cell layer, often termed endoderm (or gastrodermis) in the cnidarian literature, has a molecular profile very reminiscent of bilaterian mesoderm[5]. In line with that, the cellular behavior (partial or full EMT) during gastrulation as well as its cellular derivatives (gonads, muscles, storage tissue) is characteristic of mesoderm. Our study lends further support for the previously postulated segregation of three germ layer identities in a diploblastic animal. We also show that its early activation by MAPK is shared with mesoderm induction in many bilaterians. We therefore propose to term this tissue mesoderm in *Nematostella*.

In many bilaterians, there is an endomesodermal domain during early development, which then segregates into distinct endoderm and mesoderm[60]. In *Nematostella*, mesodermal gene expression precedes endodermal gene expression; however, we observe a transient expression of endodermal genes in the same cells[18] (this study) before it moves out into a ring of definitive endodermal tissue surrounding the mesoderm. The competence of the early mesodermal cells to express endodermal markers suggests the existence of a transient endomesodermal state in *Nematostella*.

Endodermal gene expression is rapidly suppressed in the mesoderm, while definitive endoderm is induced on the ectodermal side of the Delta/Notch boundary. This is in line with our single-cell RNAseq data, which show a stronger link of endodermal identity with the rest of the ectoderm, while mesoderm is already very distinct. Moreover, this tissue and its derivative, the septal filament, also produce cnidocytes, which typically only occur in the ectoderm. Thus, the endodermal cell fate appears to retain some features typical of ectoderm. Similarly, there are examples, in particular among the ecdysozoans, where fore- and hindgut derive from ectoderm[61,62] and continue to display some ectodermal features, such as chitinization of the epithelial layer. Thus, despite its diploblastic nature, *Nematostella* uses conserved signaling mechanisms to segregate three germ layers as many bilaterians. Preliminary evidence suggests that this is also shared in members of the other major branch of cnidarians, the scyphozoans[5], but further studies are required to confirm this.

Taken together, we find that during germ layer specification in the sea anemone *Nematostella vectensis*, MAPK signaling in a β-catenin-negative background specifies the mesodermal fate and represses the ectodermal fate, while Notch signaling induces endodermal fate in the ectodermal cells bordering the mesoderm. We propose that the involvement of MAPK signaling in mesoderm formation represents an ancestral trait shared between Cnidaria and Bilateria. In contrast, although Notch and Delta are recurrently involved in endo- and mesoderm formation in different models, a wider phylogenetic sampling is necessary to determine whether Delta/Notch-dependent endoderm specification is an ancestral feature for Cnidaria+Bilateria.

## Methods

### Animal culture and spawning

Adult polyps of *Nematostella vectensis* were cultivated in 16‰ Red Sea Salt artificial seawater (*Nematostella* medium=NM) at 18 °C in the dark, according to the established protocol[63]. Animals were fed 5x per week with freshly hatched *Artemia* nauplii. Spawning in mature polyps was induced by exposure to light over 9 h at 25 °C. After fertilization, the jelly surrounding the eggs was removed with a 3% L-cystein/NM solution for 30 min, and the embryos were washed 5−6 times in NM and raised until the required stage at 21 °C as previously described[63,64].

## Generation of single cell transcriptomic data

Embryos were collected in Eppendorf tubes pre-coated with 0.1% PBST and washed with NM (1x for 12 hpf, 2x for 8 and 10 hpf). For cell dissociation at 8 and 10 hpf, embryos were washed 3 times with 200 μL NM containing 2% L-cystein (Millipore, 52-90-4), pH 7.5. Then embryos were kept at 4 °C on ice and mixed every 3 min for 15 min. Cell suspension was centrifuged at 400 rcf for 30 s, and L-Cystein solution was removed. Then cells were resuspended in 0.5% BSA/1xPBS by pipetting. This step was repeated 2 times. For cell dissociation at 12 hpf, embryos were washed with NM and first incubated for 1 min in 5x TrypLE (diluted in NM). Dissociation was achieved by gentle pipetting for 25 min and stopped by adding an equal volume of 1% BSA/1xPBS. Then, the cell suspension was centrifuged (200 rcf, 4 min), washed with 0.5% BSA/1xPBS, and passed through a Flowmi® filter (40 μm). After dissociation and washing, the concentration and viability of all cell suspensions were determined by staining cells with ViaStain AOPI.

## Analysis of single-cell transcriptomic data

Reads were aligned to the *Nematostella vectensis* genome[65] using Cell Ranger v7.1.0[66] using standard parameters. The filtered count matrix was uploaded to Seurat v5[67–69]. Each library was filtered and normalized separately, removing cells with low read counts and low feature counts. Cell cycle markers were also regressed out using the Cell Cycle Scoring method, and new PCA and UMAP values were calculated. A detailed script can be found in our GitHub (https://github.com/technau/NemVecEndoderm) for replication of our results.

## Signaling pathway inhibitor treatments

All stock solutions of pharmacological inhibitors were prepared in DMSO. Before the first cell division, dejellied zygotes were transferred in a Petri dish with the appropriate concentration of inhibitor in NM. Zygotes were treated with a selective inhibitor of MEK (U0126, Sigma, 19−147) at 20 μM to inhibit the MAP kinase signaling, a selective inhibitor of GSK3β (1-azakenpaullone (AZ), Sigma, A3734) at 10 μM to stabilize β-catenin, and with a selective inhibitor of γ-secretase (LY411575, MedChemExpress, HY-50752) at 10−40 μM to inhibit Notch signaling.

## Microinjection of morpholinos, plasmid, and shRNA

For knockdown experiments, a previously published antisense translation blocking Nvβ-catenin morpholino (MO) (Gene Tools Inc., USA) was microinjected in zygotes at a concentration of 500 μM with 0.125 mg/ml of fluorescent Dextran-AlexaFluor488 (5′UTR Nvβ-catenin MO: 5′ TTCTTCGACTTTAAATCCAACTTCA 3′)[21]. As a control, a previously used standard morpholino known to have no effect on the development was injected at 500 μM (standard MO 5′ GATGTGCCTAGGGTACAACAACAAT 3′) as previously described[13,17]. For mosaic overexpression of the Notch intracellular domain (NICD), injection of *TBP::NICD-mCherry* plasmid into zygotes was performed as previously described[37]. To generate this vector, the cDNA fragment coding for the NICD was amplified with the insertion of an ATG at the beginning. This sequence was fused to the mCherry sequence with a 4xGly-1xSer linker (5′GGTGGTGGTGGTAGT′3) and inserted into the pJET1.2 plasmid downstream of a T7 RNA polymerase site by using NEBuilder HIFI DNA Assembly Cloning Kit (NEB; E5520S). Gene knockdown mediated by shRNA was performed as described previously[70]. ShRNA was used at 500 ng/μL. shRNA against mOrange was used as a control.

## Antibody and phalloidin staining

For immunostaining, embryos were fixed for 1 h with 4% PFA/PBST (4% paraformaldehyde dissolved in 1xPBS, 0.4% Tween 20) at room temperature (RT) without agitation and washed 4 times with PBST. Then, embryos were incubated in blocking solution (5% sheep serum, 1% bovine serum albumin (BSA) in PBST for 1 h at room temperature. Embryos were incubated overnight at 4 °C in anti-phospho-ERK (Cell

Signaling Technology, 4370S, 1:500) or anti-NICD (Cell Signaling Technology, 4147T, 1:500) diluted in blocking solution. After washing 7 times for 10 min with PBST, embryos were incubated in the blocking solution with anti-rabbit AP (Invitrogen, 31346, 1:7000) overnight at 4 °C. After washing 10 times for 10 min, staining was revealed with NBT/BCIP in AP buffer. For phalloidin staining, after fixation in 4% PFA and washing with PBST, embryos were incubated in 1 ml acetone on ice for 5 min. Specimens were washed 5 times with PBST, then incubated with 100 μL PBST containing 4 U/mL Phalloidin Alexa Fluor 488 (Thermo Fisher Scientific) and 5 μg/mL DAPI in the dark for 1 h at RT. Then, embryos were washed 7 times for 10 min with PBST, followed by infiltration with antifade mounting medium (Vectashield). Imaging was performed with a Nikon Eclipse 80i or Leica SP8 CLSM.

## Western Blot

To generate lysates, embryos were incubated in 40 μl of cell extraction buffer (ThermoFisher, FNN0011) supplemented with protease inhibitor (Roche). Samples were centrifuged at 16000 g for 10 min at 4 °C, and supernatants were resuspended in 10 μl loading dye. After gel electrophoresis and blotting, the membranes were blocked in 5% skimmed milk powder in PTw (1xPBS, 0.1% Tween). Antibodies against ERK (Cell Signaling Technology, 4695), phospho-ERK (Cell Signaling Technology, 4370), and β-actin (Cell Signaling Technology, 4970S) were diluted at 1:10,000 in blocking solution and incubated with Nitrocellulose membrane overnight at 4 °C. After washing with PTw (1x PBS, 0.1% Tween20), membranes were incubated with 1:100,000 anti-rabbit IgG conjugated to horseradish peroxidase (Sigma-Aldrich, A0545), washed with PTw, and peroxidase activity associated with proteins of interest was detected with SuperSignal West Femto Maximum Sensitivity Substrate (Thermo Fisher, 34094). An uncropped scan of the Western Blot can be found in the Source Data file.

## Cloning and in situ hybridization

To generate probes for in situ hybridization, fragments of interest were amplified from cDNA using primers listed in Supplementary Table 1. PCR products were cloned into the pGEM-T vector (Promega, A3600). FITC or DIG-labeled RNA probes were generated by in vitro transcription with SP6 or T7 polymerase. In situ hybridization was carried out as previously described[17] with some modifications.

After incubation with anti-Digoxigenin AP (Roche, 11093274910, 1:4000), embryos were washed 10 times 10 min with TBST (0.4% Tween). Background was removed by washing embryos 6 times with TBST (0.4% Tween) and kept overnight at 4 °C. Two-color fluorescent in situ hybridization was performed similarly to single in situ hybridization according to the protocol described[71]. Fluorescent staining was revealed with the TSA Plus Fluorescein and TSA Plus Cy3 detection Kits (AKOYA Biosciences). After staining, embryos were infiltrated with Vectashield (VectorLabs) and imaged using Leica SP8 CLSM.

## Statistics and Reproducibility

In this study, no data were excluded. All experiments were performed on at least three independent biological replicates. In general, for each condition, a minimum of 30 embryos with an average sample size of 50 embryos were processed per replicate based on established practices in the field. No Statistical methods were used to determine a significant difference between groups.

## Reporting summary

Further information on research design is available in the Nature Portfolio Reporting Summary linked to this article.

# Data availability

The single-cell data generated in this study have been deposited in the GEO database under accession code GSE302686. Source data are provided with this paper.

## Code availability

A detailed script for the analysis of the single-cell RNAseq data can be found on our GitHub (https://github.com/technau/NemVecEndoderm).

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

## Acknowledgements

This research was funded in whole or in part by the Austrian Science Fund (FWF) grants to U.T. (P34404) and Austrian Science Foundation (FWF) grants 10.55776/P30404 and 10.55776/P36080 to G.G. T.L. was a recipient of the fellowship ICM-2017-07957 of the Stipendienstiftung der Republik Österreich. For the purpose of Open Access, the authors have applied a CC BY public copyright license to any Author Accepted Manuscript (AAM) version arising from this submission.

## Author contributions

E.H. conceived and carried out most of the experiments. He wrote the initial draft of the paper. J.S. generated the single-cell transcriptome, J.D.M. and A.G.C. performed the bioinformatic and statistical analyses of the transcriptome data, GG and TL carried out Western blot and electron microscopy, U.T. and E.H. conceived the study, analyzed the data, and wrote the paper. All co-authors edited the paper.

## Competing interests

The authors declare no competing interests.
