## [Peer Review file · Nature Communications]

Segregation of endoderm and mesoderm germ layer identities in the diploblast *Nematostella vectensis*

Corresponding Author: Professor Ulrich Technau

Version 0:

Reviewer comments:

Reviewer #1

(Remarks to the Author)

The manuscript by Haillot et al. presents an in-depth study of germ layer specification in *Nematostella vectensis*, focusing on the interplay between the β -catenin, MAPK, and Notch signaling pathways. The authors provide compelling evidence that mesoderm is specified early via MAPK signaling, that nuclear β -catenin promotes ectodermal and later endodermal identity, and that Notch signaling is both necessary and sufficient for endoderm formation at the mesoderm-ectoderm boundary. The study challenges traditional views of cnidarian germ layers and suggests that the genetic mechanisms underlying triploblasty may have originated prior to the Bilateria-Cnidaria split.

The study is well-conducted, with a robust in situ hybridization screen, single-cell transcriptomic analysis, functional perturbations, and molecular assays. The conclusions are well-supported by the data, and the findings are relevant to developmental and evolutionary biology. I believe this study will be of broad interest to the readership of Nature Communications and am in favor of its publication. However, I believe there are several issues that should be addressed prior to publication:

- These data indicate that β -catenin suppresses mesoderm and promotes ectoderm and later endoderm. However, some previous studies suggested a pro-mesodermal role for β -catenin in *Nematostella*. A short discussion of this discrepancy would be useful.
- Can the authors speculate on what upstream signals activate MAPK in the mesoderm? This would be a valuable addition to the discussion.
- If feasible, in figure 1 it would be very satisfying and convincing to show a double fluorescent in situ for a mesodermal marker and an endodermal marker to demonstrate that there is little or no spatial overlap. Alternatively, this could be shown in Fig. 5. Either way, it would support the model.
- The Cadherin staining in Fig. 4B is difficult to see because the micrographs are small. These would be better presented as larger images, as is done in Supp. Fig. 4. At a minimum, an inset showing the ectopic localization in the inhibitor treatment at higher resolution could be shown to aid the reader.
- In Fig. 4C and 4D, could the color overlay be in a brighter / more visible color? I find the dark blue to be more difficult to see in these panels than in 4B. Perhaps red or orange?
- The authors mention DAPT treatments in the text, but as far as I can see, none of the data in the figures is labeled as DAPT. I suggest that the reference to DAPT be removed for clarity, or, if there is additional DAPT data, it could be added as supplemental.
- The red-green color palette in Figure 5B and 5C is difficult to distinguish for readers with color blindness. I recommend changing one of the colors, for instance swapping red with magenta, to make this figure more accessible.
- In Supp. Fig. 6 the color scheme for the ectodermal domains in the model shown on the right is switched relative to the models shown in Fig. 4F and Fig. 7. I suggest switching the color scheme in Supp Fig. 6 to be consistent with the rest.

Reviewer #2

(Remarks to the Author)

The manuscript "Notch, β -catenin and MAPK signaling segregate endoderm and mesoderm germ layer identities in the diploblast *Nematostella vectensis*" by Haillot et al. presents genetic evidence for the presence of a third germ layer,

mesoderm, in the developing *Nematostella* embryo. To support its existence, they present a time course of early expression of various genes associated with ectoderm, mesoderm and endoderm identity. Part of the categorization of these identities is based on orthologous gene expression in Bilateria and expression similarities during gastrulation in echinoderms. The authors functionally test the establishment of these various markers and their germ layer derivatives using chemical antagonists and genetic knockdowns of the β -catenin/Wnt, MAPK and Notch signaling pathways. From these manipulations, their main findings are that early β -catenin signaling represses mesoderm while promoting ectoderm, MAPK signaling is required for specification of the mesoderm, and that the regulation of Notch signaling at the border between ectoderm and mesoderm is required for the induction of endoderm formation.

Scientifically, the primary claims of the paper are well supported. The level of analysis and results from knockdowns and chemical antagonist experiments supports the authors' claims that the early *Nematostella* embryo prepatterns groups of cells, prior to gastrulation, to specify layers with unique identities that resemble their bilaterian germ layer counterparts, specifically mesoderm and endoderm. However, there are two main points of concern which the authors must address before the manuscript is fully accepted for publication. The first concern is the small size/resolution of several panels and graphics in the figures (which is further detailed under Major Points). This small formatting makes it hard to discern details that are described in the main body of the text. The second major concern is the duplication of data figure (Figure 4B supplementary, the same image is used for the oral view of brachyury) -- this is most likely an oversight, but we would ask that the authors carefully review all figures and legends to ensure that this type of error hasn't been duplicated.

Major Points:

1. The large number of in situ panels, while informative, makes several of the figures difficult to read and interpret. In particular, the time courses and graphics, which are helpful, are too small and hard to read clearly. This includes the timelines in figures 2B and 3C and the graphics in figures 4F and 5E.
2. Similarly, the data in figure 6 needs to be re-organized so that the data panels are larger. I can't see any difference in the LY-4111575 treated embryos in panel 6C because it is too small.
3. The inference from the single cell RNAseq UMAP "to distinguish cell populations with ectodermal, mesodermal and endodermal identities" is not well supported. We can clearly see areas on the UMAP where clusters of cells are simultaneously expressing all three (germ layer) markers. Furthermore, the authors state that "at 8 hpf, only clusters of mesodermal and ectodermal cells are clearly identifiable" yet brachyury, which they are using to mark (presumptive?) endoderm, is clearly expressed at 8 hpf. The complementary in situ patterns do appear to support a specification of mesoderm and endoderm fates, but there appears to be significant overlap which is supported by the UMAP pattern. The development of *Nematostella* is regulative and could account for the overlap of germ layer markers at this early stage, but nowhere in the text are these observations addressed (except briefly in the discussion).
4. Figure 2 deals with the timing of germ layer specification as a result of treating the embryos with the Wnt/ β -catenin signaling inhibitor, azakenpaullone (AZ) at either 8 or 20 hpf. From these experiments, the authors propose that β -catenin signaling promotes endodermal fate at later stages (as shown by the expansion of endodermal markers at 20 hpf in the AZ treated embryos). An additional panel for ectodermal markers at 20 hpf of AZ treated embryos would provide insight as to whether the embryos ubiquitously express both endodermal and ectodermal markers or solely endodermal markers.
5. Figure 4: The morphological data supporting the role of MAPK signaling in mesoderm formation isn't clear (from the images). While the U0126 treated embryos clearly fail at gastrulation, it is hard to tell what is happening in figures 4C and 4D. Potentially dotted outlines or arrowheads would help draw the reader's attention to the cellular defects in the treated embryos which aren't obvious with the figures at their current state of resolution.
6. Figure 4B Supplementary: The images for brachyury, shcontrol OV and shERG LV appear to be duplications of the same image.

Minor Points:

Line 57: Fix sentence "...mesoderm is one the first cell fate..." to "...mesoderm is one of the first cell fate..."

Line 68: anemone is misspelled.

Line 187: The authors say that expression of hmx2 is detectable by 16 hpf, but the image in Figure 1B does not show any expression at this time point.

Figure 1: What is the difference between SV (side view) and LV (lateral view)? They appear to be the same. Also the abbreviation for LV (lateral view) is not defined in the figure's legend.

Figure 1D. For the general audience, viewpoint references should be added either to the figure or mentioned in its legend. i.e., the top row of embryos is presented in the oral view, whereas the second row is the lateral view.

Line 193: Please remove the second "the" from the following sentence "...surrounding the mesoderm followed by the wnt1 and wnt3 gene expression", or rewrite the sentence as "Within the broad ring of brachyury expression, smaller rings of wnt1

and wnt3 expression appear.”

Line 282: The figure reference of 3D isn't correct. What the authors are referring to is the data in Figure 3E.

Reviewer #3

(Remarks to the Author)

This manuscript marshals many different observations and experiments (including single cell sequencing at distinct early developmental time points, in situ hybridizations, western blots, pharmacological treatments, and knockdowns) to develop a very compelling model for germ line specification. This model is clearly articulated right in the abstract:

- "the mesodermal territory is specified at the animal pole at 6 hours postfertilization, followed by the specification of the definitive endoderm between mesoderm and ectoderm"

- "mesodermal marker genes are activated by MAPK signaling while being repressed elsewhere by β -catenin signaling"

- "Delta-expressing mesoderm then signals to Notch-expressing ectoderm inducing the definitive endoderm domain at the mesoderm/ectoderm interface."

Given these results, the authors then hypothesize that "Given the similarity of the germ layer specification between the sea anemone and echinoderms, we propose that triploblasty may have predated the split of cnidarians and bilaterians".

This work builds directly on Steinmetz et al 2017 (which shares authors with this manuscript), and is in many ways a direct follow-up. In that earlier paper, they argued for a radical reassessment of the homology of cnidarian germ layers. In their scheme:

Old term -> Their new term

Ectoderm -> Ectoderm

Endoderm -> Mesoderm

Pharyngeal ectoderm -> Endoderm

They argue that the new results support this reassessment of homology, and therefore the corresponding terminology update.

The work is beautifully executed, thorough, and clearly presented. Without doubt the results will be of great interest and serve as a productive foundation for future work. It makes important advances in the mechanisms and details of germ line specification in a cnidarian.

I was, though, and still remain after reading this exciting new manuscript, skeptical of their specific reinterpretation of germ layer homology. My concern is with the scope of the question. They are asking "Which cnidarian germ layers are homologous to which bilaterian germ layers?" Within this scope, evidence is weighed to argue for and against different 1:1 mappings across evolution. But this presupposes that such a clean 1:1 mapping exists. One could also interpret their results as arguing that there is not a clean 1:1 mapping between germ layers, which is also an exciting and fascinating result. Maybe the most recent common ancestor did not even have germ layers that clearly correspond to all those seen in any one extant animal. Without being open to this possibility, the excellent data being generated may lead to the wrong conclusions. A presumed 1:1 mapping will also make it more difficult to expand the conceptual framework developed here to other animal groups, specifically ctenophores, sponges, and placozoans.

This intellectual disagreement is GOOD, and is possible because of the excellent results presented here. In no way am I arguing against publication based on disagreements in interpretation, I would just encourage the authors to word the discussion in particular in a way that opens the door to these broader discussions. But such a decision is theirs.

Minor comments:

- line 21: "investigated, how" strike the comma

- line 13: "genes, whose" strike the comma

Version 1:

Reviewer comments:

Reviewer #1

(Remarks to the Author)

The authors have been careful in taking all the reviewers' recommendations and suggestions into account and have added

a number of experiments and figures to the manuscript. This well-executed study, in my view, should now be published, it will be an excellent addition to and stimulate new thinking in the evo-devo field.

Reviewer #2

(Remarks to the Author)

The authors have fully addressed the concerns raised in our review. This is an excellent piece of work with deep implications for understanding the evolution of animal development.

REVIEWER COMMENTS

Reviewer #1 (Remarks to the Author):

The manuscript by Haillot et al. presents an in-depth study of germ layer specification in *Nematostella vectensis*, focusing on the interplay between the β -catenin, MAPK, and Notch signaling pathways. The authors provide compelling evidence that mesoderm is specified early via MAPK signaling, that nuclear β -catenin promotes ectodermal and later endodermal identity, and that Notch signaling is both necessary and sufficient for endoderm formation at the mesoderm-ectoderm boundary. The study challenges traditional views of cnidarian germ layers and suggests that the genetic mechanisms underlying triploblasty may have originated prior to the Bilateria-Cnidaria split. The study is well-conducted, with a robust in situ hybridization screen, single-cell transcriptomic analysis, functional perturbations, and molecular assays. The conclusions are well-supported by the data, and the findings are relevant to developmental and evolutionary biology. I believe this study will be of broad interest to the readership of Nature Communications and a min favor of its publication. However, I believe there are several issues that should be addressed prior to publication:

- These data indicate that β -catenin suppresses mesoderm and promotes ectoderm and later endoderm. However, some previous studies suggested a pro-mesodermal role for β -catenin in *Nematostella*. A short discussion of this discrepancy would be useful.

An additional paragraph in the Discussion has been added to explain the difference between our results and some previous studies.

- Can the authors speculate on what upstream signals activate MAPK in the mesoderm? This would be a valuable addition to the discussion.

We thank the reviewer for this interesting suggestion. We now added new data (new Sup Fig. 3) addressing the upstream regulation of the MAPK signaling pathway in the mesoderm. We investigated the potential role of Mos kinases, which are known to directly phosphorylate MEK independently of ligand-receptor interactions and to activate the MAPK signaling pathway. We searched for Mos homologs expressed during early mesoderm specification in the single cell dataset. We identified *mos2* as a candidate gene and experimentally showed that *mos2* is indeed promoting mesoderm specification upstream of MAPK signaling.

- If feasible, in figure 1 it would be very satisfying and convincing to show a double fluorescent in situ for a mesodermal marker and an endodermal marker to demonstrate that there is little or no spatial overlap. Alternatively, this could be shown in Fig. 5. Either way, it would support the model.

Thanks for this suggestion. To support the model as you mentioned, we have added a double fluorescent in situ showing the complementary expression of *delta* and *foxA* in the Figure 5. *delta* is expressed in the mesoderm (magenta) and *foxA* (yellow) in the endoderm surrounding the mesoderm.

- The Cadherin staining in Fig. 4B is difficult to see because the micrographs are small. These would be better presented as larger images, as is done in Supp. Fig. 4. At a minimum, an inset showing the ectopic localization in the inhibitor treatment at higher resolution could be shown to aid the reader.

To address this point, we divided the Figure 4 into 2 separate figures (Figure 4 and Figure 5). The first figure illustrates the essential role of MAPK signaling in the mesoderm, while the second figure demonstrates the inhibitory roles of β -catenin and MAPK signaling on aboral ectoderm identity. With this new configuration, we were able to include a larger image in 4B. We also added a close-up image with red asterisks to visualize the ectopic localization of Cadherin 3 in the mesoderm.

- In Fig. 4C and 4D, could the color overlay be in a brighter / more visible color? I find the dark blue to be more difficult to see in these panels than in 4B. Perhaps red or orange?

Thanks for this comment. The dark blue color was changed with magenta color to improve visibility and to maintain consistency with the color used for the mesoderm in general in the other figures.

- The authors mention DAPT treatments in the text, but as far as I can see, none of the data in the figures is labeled as DAPT. I suggest that the reference to DAPT be removed for clarity, or, if there is additional DAPT data, it could be added as supplemental.

In fact, we have initially carried out treatments also with DAPT, with essentially the same result as LY411575. However, the latter drug turned out more efficient, therefore we continued with this drug for all following experiments. To avoid confusion, we have now removed DAPT from the text.

- The red-green color palette in Figure 5B and 5C is difficult to distinguish for readers with color blindness. I recommend changing one of the colors, for instance swapping red with magenta, to make this figure more accessible.

As you suggested, we have changed the colors to be accessible for readers with color blindness. Green has been replaced with cyan, and red has been changed to magenta.

- In Supp. Fig. 6 the color scheme for the ectodermal domains in the model shown on the right is switched relative to the models shown in Fig. 4F and Fig. 7. I suggest switching the color scheme in Supp Fig. 6 to be consistent with the rest.

We have updated the colors in Supplementary Figure 6 to assure consistency with those used in the other schemas.

Reviewer #2 (Remarks to the Author): The manuscript "Notch, β -catenin and MAPK signaling segregate endoderm and mesoderm germ layer identities in the diploblast *Nematostella vectensis*" by Haillet et al. presents genetic evidence for the presence of a third germ layer, mesoderm, in the developing *Nematostella* embryo. To support its existence, they present a time course of early expression of various genes associated with ectoderm, mesoderm and endoderm identity. Part of the categorization of these identities is based on orthologous gene expression in Bilateria and expression similarities during gastrulation in echinoderms. The authors functionally test the establishment of these various markers and their germ layer derivatives using chemical antagonists and genetic knockdowns of the β -catenin/Wnt, MAPK and Notch signaling pathways. From these manipulations, their main findings are that early β -catenin signaling represses mesoderm while promoting ectoderm, MAPK signaling is required for specification of the mesoderm, and that the regulation of Notch signaling at the border between ectoderm and mesoderm is required for the induction of endoderm formation. Scientifically, the primary claims of the paper are well supported. The level of analysis and results from knockdowns and chemical antagonist experiments supports the authors' claims that the early *Nematostella* embryo prepatterning groups of cells, prior to gastrulation, to specify layers with unique identities that resemble their bilaterian germ layer counterparts, specifically mesoderm and endoderm. However, there are two main points of concern which the authors must address before the manuscript is fully accepted for publication. The first concern is the small size/resolution of several panels and graphics in the figures (which is further detailed under Major Points). This small formatting makes it hard to discern details that are described in the main body of the text. The second major concern is the duplication of data figure (Figure 4B supplementary, the same image is used for the oral view of brachyury) -- this is most likely an oversight, but we would ask that the authors carefully review all figures and legends to ensure that this type of error hasn't been duplicated. Major Points:

1. The large number of in situ panels, while informative, makes several of the figures difficult to read and interpret. In particular, the time courses and graphics, which are helpful, are too small and hard to read clearly. This includes the timelines in figures 2B and 3C and the graphics in figures 4F and 5E.

Because the figures were included in the text at a medium to low resolution in the initial submission, this may have led to the problem of legibility. To account for that, we have reorganized the original Figures 2, 3, 4 and 5 to make the data panels bigger and easier to read. We hope that they are legible now.

2. Similarly, the data in figure 6 needs to be re-organized so that the data panels are larger. I can't see any difference in the LY-4111575 treated embryos in panel 6C because it is too small.

To address this problem, we also reorganized Figure 6 to make panel 6C larger.

3. The inference from the single cell RNAseq UMAP "to distinguish cell populations with ectodermal, mesodermal and endodermal identities" is not well supported. We can clearly see areas on the UMAP where clusters of cells are simultaneously expressing all three (germ layer) markers. Furthermore, the authors state that "at 8 hpf, only clusters of mesodermal and ectodermal cells are clearly identifiable" yet brachyury, which they are using to mark (presumptive?) endoderm, is clearly expressed at 8 hpf. The complementary in situ patterns do appear to support a specification of mesoderm and endoderm fates, but there appears to be significant overlap which is supported by the UMAP pattern. The development of *Nematostella* is regulative and could account for the overlap of germ layer markers at this early stage, but nowhere in the text are these observations addressed (except briefly in the discussion).

We thank the reviewer for this point. Indeed, *brachyury* expression is more dynamic than other typical endodermal markers such as *foxA*, *wnt1*, *wnt3*. The onset of *brachyury* expression is very early (at least at 8hr) in a broader domain of the oral ectodermal half, yet already excluding the future mesodermal plate at 10hr. We assume that this early activation is due to early maternal β -catenin (Lebedeva et al., 2025). The segregation of mesoderm and endoderm and hence activation of Notch signaling at the border further strengthens and confines brachyury expression in the future endoderm, overlapping with *foxA* and other genes. Yet, the border between endoderm and ectoderm is less strictly defined due to the contiguous gradient of *bra* expression. Thus, *brachyury* should not be considered as a strict endodermal marker because it is also expressed (at much lower level) in the oral midbody ectoderm during blastula and early gastrula. This may indeed reflect the regulative development of this organism and is also corroborated by the

UMAP of the single cell data. This is why we prefer to use *foxA* as a strict endodermal marker. A description of this has been added to the text to clarify its earlier profile.

4. Figure 2 deals with the timing of germ layer specification as a result of treating the embryos with the Wnt/ β -catenin signaling inhibitor, Azakenpaullone (AZ) at either 8 or 20 hpf. From these experiments, the authors propose that β -catenin signaling promotes endodermal fate at later stages (as shown by the expansion of endodermal markers at 20 hpf in the AZ treated embryos). An additional panel for ectodermal markers at 20 hpf of AZ treated embryos would provide insight as to whether the embryos ubiquitously express both endodermal and ectodermal markers or solely endodermal markers.

To be consistent with the model we propose, 2 ectodermal genes (*six3/6* and *foxQ2a*) with aboral expression have been added to Figure 2 at 20hpf.

5. Figure 4: The morphological data supporting the role of MAPK signaling in mesoderm formation isn't clear (from the images). While the U0126 treated embryos clearly fail at gastrulation, it is hard to tell what is happening in figures 4C and 4D. Potentially dotted outlines or arrowheads would help draw the reader's attention to the cellular defects in the treated embryos which aren't obvious with the figures at their current state of resolution.

To clarify the morphological effect of U0126 treatment on embryos during gastrulation in Figure 4C and 4D, the dark blue color was replaced with magenta. We also added several arrowheads in these figures to indicate the absence of filopodia in U0126-treated embryos.

6. Figure 4B Supplementary: The images for brachyury, shcontrol OV and shERG LV appear to be duplications of the same image.

Thank you for spotting this mistake. We have replaced the image with the correct one corresponding to the experiment.

Minor Points: Line 57: Fix sentence "...mesoderm is one the first cell fate..." to "...mesoderm is one of the first cell fate..."

This point has been revised in the text.

Line 68: anemone is misspelled.

This point has been modified in the text.

Line 187: The authors say that expression of *hmx2* is detectable by 16 hpf, but the image in Figure 1B does not show any expression at this time point.

We apologize for this and have added a sentence to clarify the beginning of *hmx2* expression at 18hpf in the text to align with the data shown in the Figure 1B.

Figure 1: What is the difference between SV (side view) and LV (lateral view)? They appear to be the same. Also the abbreviation for LV (lateral view) is not defined in the figure's legend.

We have added definitions of these two terms in the text. When the orientation along the oral-aboral axis is not clear, we use the term "side view". When we have sufficient information to identify the oral aboral axis, we use the term "lateral view".

Figure 1D. For the general audience, viewpoint references should be added either to the figure or mentioned in its legend. i.e., the top row of embryos is presented in the oral view, whereas the second row is the lateral view.

We have added the viewpoint reference directly to the figure.

Line 193: Please remove the second "the" from the following sentence "...surrounding the mesoderm followed by the *wnt1* and *wnt3* gene expression", or rewrite the sentence as "Within the broad ring of brachyury expression, smaller rings of *wnt1* and *wnt3* expression appear."

The sentence was revised according to the recommendation.

Line 282: The figure reference of 3D isn't correct. What the authors are referring to is the data in Figure 3E.

It has been modified.

Reviewer #3 (Remarks to the Author):

This manuscript marshals many different observations and experiments (including single cell sequencing at distinct early developmental time points, in situ hybridizations, western blots, pharmacological treatments, and knockdowns) to develop a very compelling model for germ line specification. This model is clearly articulated right in the abstract: - "the mesodermal territory is specified at the animal pole at 6 hours post fertilization, followed by the specification of the definitive endoderm between mesoderm and ectoderm" - "mesodermal marker genes are activated by MAPK signaling while being repressed elsewhere by β -catenin signaling" - "Delta-expressing mesoderm then signals to Notch-expressing ectoderm inducing the definitive endoderm domain at the mesoderm/ectoderm interface." Given

these results, the authors then hypothesize that "Given the similarity of the germ layer specification between the sea anemone and echinoderms, we propose that triploblasty may have predated the split of cnidarians and bilaterians". This work builds directly on Steinmetz et al 2017 (which shares authors with this manuscript), and is in many ways a direct follow-up. In that earlier paper, they argued for a radical reassessment of the homology of cnidarian germ layers. In their scheme:

Old term -> Their new term

Ectoderm -> Ectoderm

Endoderm -> Mesoderm

Pharyngeal ectoderm -> Endoderm

They argue that the new results support this reassessment of homology, and therefore the corresponding terminology update. The work is beautifully executed, thorough, and clearly presented. Without doubt the results will be of great interest and serve as a productive foundation for future work. It makes important advances in the mechanisms and details of germ line specification in a cnidarian. I was, though, and still remain after reading this exciting new manuscript, skeptical of their specific reinterpretation of germ layer homology. My concern is with the scope of the question. They are asking "Which cnidarian germ layers are homologous to which bilaterian germ layers?" Within this scope, evidence is weighed to argue for and against different 1:1 mappings across evolution. But this presupposes that such a clean 1:1 mapping exists. One could also interpret their results as arguing that there is not a clean 1:1 mapping between germ layers, which is also an exciting and fascinating result. Maybe the most recent common ancestor did not even have germ layers that clearly correspond to all those seen in any one extant animal. Without being open to this possibility, the excellent data being generated may lead to the wrong conclusions. A presumed 1:1 mapping will also make it more difficult to expand the conceptual framework developed here to other animal groups, specifically ctenophores, sponges, and placozoans.

This intellectual disagreement is GOOD, and is possible because of the excellent results presented here. In no way am I arguing against publication based on disagreements in interpretation, I would just encourage the authors to word the discussion in particular in a way that opens the door to these broader discussions. But such a decision is theirs.

We thank the reviewer for this positive and inspiring comments. Indeed, we do not intend to have the final word on the discussion of the homology of germ layers. Please note that we did not show a final figure with a phylogenetic tree, but just show the comparison of two species, which we consider compelling though. We feel that on the basis of the existing data, our proposed scenario appears the most plausible one at this point. However, similar analyses should be carried out in several more species to come to a final conclusion.

Minor comments:

- line 21: "investigated, how" strike the comma

The comma has been removed.

- line 13: "genes, whose" strike the comma

The comma has been removed.